# A reversibly gated protein-transporting membrane channel made of DNA

Swarup Dey [1,2,5], Adam Dorey [3,5], Leeza Abraham [1,2,5], Yongzheng Xing [3], Irene Zhang[1], Fei Zhang [4], Stefan Howorka [3,6✉] & Hao Yan [1,2,6✉]

Controlled transport of biomolecules across lipid bilayer membranes is of profound significance in biological processes. In cells, cargo exchange is mediated by dedicated channels that respond to triggers, undergo a nanomechanical change to reversibly open, and thus regulate cargo flux. Replicating these processes with simple yet programmable chemical means is of fundamental scientific interest. Artificial systems that go beyond nature's remit in transport control and cargo are also of considerable interest for biotechnological applications but challenging to build. Here, we describe a synthetic channel that allows precisely timed, stimulus-controlled transport of folded and functional proteins across bilayer membranes. The channel is made via DNA nanotechnology design principles and features a 416 nm$^2$ opening cross-section and a nanomechanical lid which can be controllably closed and re-opened via a lock-and-key mechanism. We envision that the functional DNA device may be used in highly sensitive biosensing, drug delivery of proteins, and the creation of artificial cell networks.

[1] Biodesign Center for Molecular Design and Biomimetics (at the Biodesign Institute) at Arizona State University, Tempe, AZ 85287, USA. [2] School of Molecular Sciences, Arizona State University, Tempe, AZ 85287, USA. [3] Department of Chemistry & Institute of Structural Molecular Biology, University College London, London, UK. [4] Department of Chemistry, Rutgers University, Newark, NJ 07102, USA. [5] These authors contributed equally: Swarup Dey, Adam Dorey and Leeza Abraham. [6] These authors jointly supervised this work: Stefan Howorka, Hao Yan. ✉email: s.howorka@ucl.ac.uk; hao.yan@asu.edu

Controlled molecular transport across membranes is biologically vital as illustrated by myriads of ligand-gated channels with sophisticated architecture and defined function[1]. The channels perform a wide range of roles such as in signal transduction and amplification and import of nutrients. While varied, most channels regulate the transport of ions and small molecules by binding a ligand, which causes a nanomechanical change to alter the channels' transport properties, until the ligand dissociates. Harnessing this valve-like function is of considerable technological interest in signal-amplified point-of-care diagnostics[2], delivery of therapeutics[3,4], cell biological research[5,6], and biomimetic cell signaling[7,8]. Yet, using natural channels beyond their biogenic remit can be difficult due to not only their often-fragile protein architectures and narrow size-range for cargo, but also a limited choice of triggers, and reduced control when a channel switches back to its original state.

Building programmable synthetic channels for defined transport of large biorelevant cargo would hence be a step-change. So far, de novo design has been achieved with barrel-like and constitutively open protein pores of a few nm width[9,10]. The membrane pores are key in portable DNA sequencing[11–14] and label-free sensing[15–21] by registering molecules that pass the pore lumen[17,22–24]. Leveraging the success of nanopores for constructing proteinaceous ligand-gated channels is, however, hindered by the current challenges in integrating molecular recognition, nanomechanical change, and transport function.

Compared to other construction routes, de novo design with DNA nanotechnology[25–29] offers unprecedented structural precision and tunability[30], dynamic-nanomechanical control[31–35], a wide range of chemical modifications[36–39], and stability in harsh conditions[40–42]. Rational design with DNA has previously led to various types of membrane-spanning DNA nanopores[43–60]. Mimicking the reversible gating behavior similar to that of natural ion channels has been a long-standing desire of the field[43,48,53] due to the potential to expand the application of DNA nanopores to transport bioactive cargo into cells, or to construct cell–cell communication[61] for artificial gap junctions and to enable integration of artificial tissues with living cells for tissue engineering[62]. However, on-demand reversible gating has remained elusive, as previously demonstrated DNA nanopores are one-way and partial[47,48,53,63]—once the lid is opened it cannot be closed or vice-versa. Moreover, systematically increasing the internal pore diameter has been a constant driver in the field. Wider pore lumens would advance the application of DNA nanopores in the field of protein sensing, to bridge the gap between narrow-biological pores that cannot accommodate fully folded proteins and solid-state nanopores that lack the tunability and precision of biological pores. DNA nanopores have previously demonstrated[43,44,48,49] large molecule transport, but not in the context of a reversibly gated pore. This is due to the inherent design challenges associated with parallel lattice alignment[64] in all previous designs[43,44,48–51,55] that limits available scaffold to simultaneously accommodate a wide pore, a controllable lid, and a plate wide enough to place a large number of cholesterol moieties to balance the energetics of a wide pore insertion (more detailed discussion in the results section and Supplementary Fig. 1a).

Here, we use a "horizontal routing" based DNA origami design strategy for building a reversibly key-and-lock gated square-channel with 20.4 nm × 20.4 nm cross-sectional opening to allow precisely timed transport of folded proteins across membranes (Fig. 1a–c), which is all beyond nature's functional remit. The creation of our synthetic channel explores the wider scope of DNA nanotechnology to integrate binding of artificial ligands, triggered nanomechanical changes for opening and closing, and transport of nanoscale large cargo across membranes.

## Results

**Design of a large and gated DNA channel.** Before designing our large and gated channel (LGC) (Fig. 1a–c) we examined whether previous nanoarchitecture principles could be applied. In the previous DNA nanopores[43,44,48–51,59], duplex helices are aligned parallel in lattice fashion, as dictated by the widely used caD-NAno design software[64]. In those cases, the duplexes were routed "vertically", i.e. at 90 degrees relative to the membrane plane (Supplementary Fig. 1a). This arrangement places the majority of DNA into an extramembrane cap region with multi-duplex layer-thick channel walls while restricting the amount of DNA available for forming a wide membrane-embedded pore (Supplementary Fig. 1a). The small lateral footprint of the pore also limits the number of attachment points for lipid anchors required for efficient pore insertion into bilayer membranes. To overcome these restrictions, we opted to route the component helices 'horizontally' to the membrane by using the free-form software Tiamat[65] (Supplementary Fig. 1b, c).

Using the horizontal routing, we rationally designed the channel featuring a square-channel lumen with 20.4 nm × 20.4 nm opening cross-section into a single-duplex layer DNA origami plate of 70 nm × 70 nm external dimensions (Fig. 1a–c). The four-duplex deep channel spans the bilayer membrane while the large base plate sits on top of the bilayer (Fig. 1a). By drastically expanding the footprint of the extramembrane cap, total 160 hydrophobic cholesterol anchors can be accommodated for efficient membrane insertion, out of which, 64 cholesterol anchors were used in the current study (Fig. 1a and Supplementary Fig. 2). As a further advantage of the design, the channel with 416 nm$^2$ opening cross-section can be reversibly closed and opened with a square lid composed of horizontally routed DNA duplexes to yield the closed channel LGC-C (Fig. 1a) and the open version LGC-O (Fig. 1b). To achieve this dynamic change, the lid is attached at one side to the channel plate by flexible hinges (Supplementary Fig. 3). The other lid side carries two single-stranded half locks which can hybridize with the complementary half locks at the base plate to form complete duplex locks (Fig. 1d). To open the lock and lid of the closed channel (LGC-C), a pair of single-stranded DNA keys dissociate the locks to form the open channel (LGC-O) (Fig. 1d and Supplementary Fig. 4). The opened lid can be switched back to close by a single-stranded reverse key pair (Fig. 1d). This externally controlled mechanism is expected to reversibly switch the lid-gated channel between an open and closed state to regulate the flow of large molecular cargo.

We first assembled non-lid version LGC-N and the nanostructures LGC-C and LGC-O carrying the lid in two states, by annealing the scaffold DNA with staple oligonucleotides (Supplementary Data 1). The structures did not yet contain the cholesterol anchors. OxDNA simulations[66] showed the desirable formation of LGC-N, LGC-C, and LGC-O in solution (Supplementary Fig. 6 and Supplementary Video 1) and gel electrophoresis confirmed that single assembly products had formed (Supplementary Fig. 5). Atomic force microscopy (AFM) (Fig. 1e–g and Supplementary Fig. 6) and transmission electron microscopy (TEM) (Supplementary Fig. 7) established the expected dimensions. For example, the AFM-derived external average side-lengths of LGC-N was 78.0 ± 2.9 nm for the square plate and 22.7 ± 5.1 nm ($n = 65$) (Fig. 1e and Supplementary Fig. 6) for the square opening, close to the expected values of 70 and 20.4 nm, respectively. The elevations at the nanostructure center stem from the four-duplex-high channel walls extending from LGC bound top-down to the mica substrate. By comparison, the closed-lid LGC-C featured no central opening, and the side-lengths were 77.4 ± 2.6 nm for the plate and 22.7 ± 1.9 nm for the channel wall ($n = 11$) (Fig. 1f and Supplementary Fig. 6).

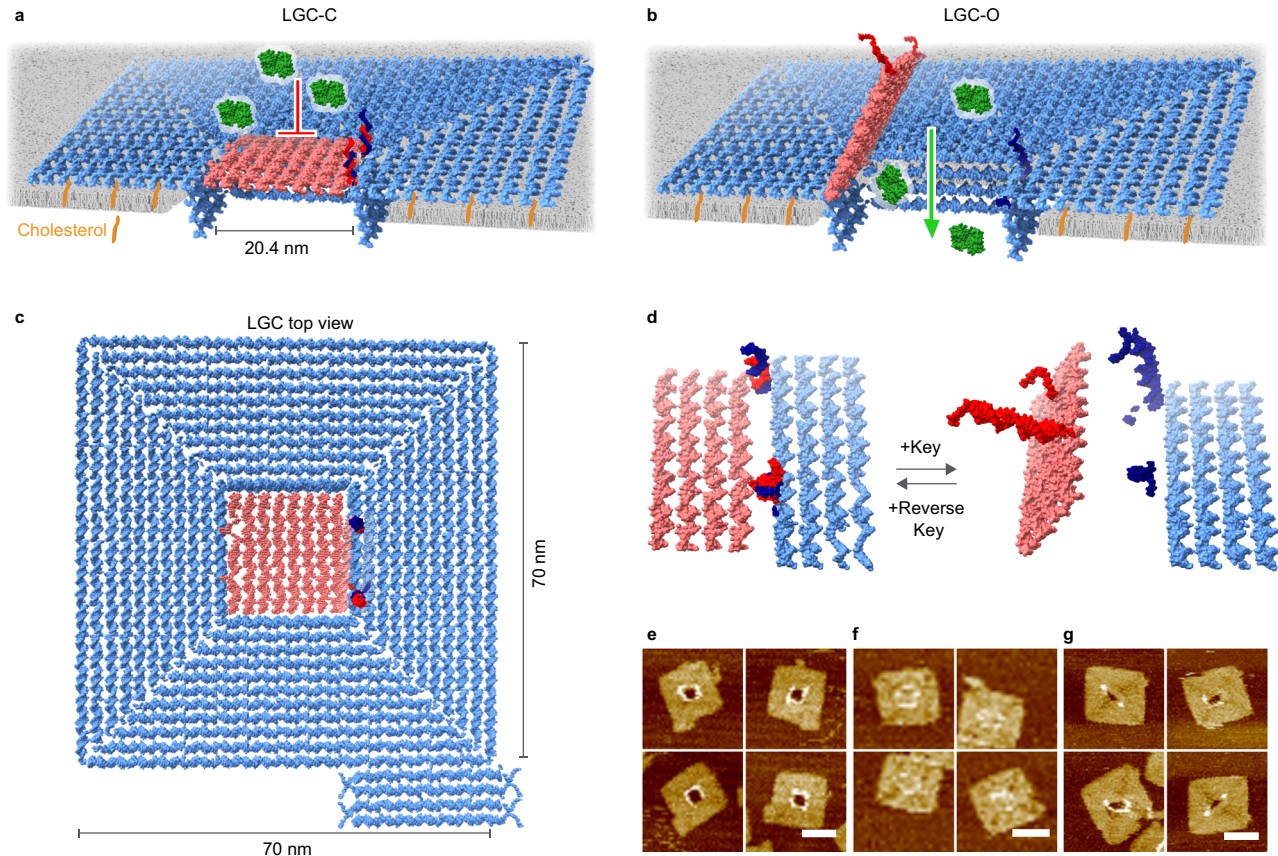

**Fig. 1 A rationally designed large and gated channel (LGC). a–c** Structural model of the large and gated channel (LGC) containing a channel lumen with 20.4 nm × 20.4 nm opening cross-section into a 70 nm × 70 nm single-duplex layer square plate (blue). A total of 64 hydrophobic cholesterol anchors (orange), placed around the pore on the bottom surface of the plate helps the nanopore to insert through the bilayer. A lid (pink) can be reversibly closed (**a**) and opened (**b**), by a key and reverse key mechanism to control the transmembrane flux of cargoes. **c** Top view of LGC. **d** Opening and closing mechanism with key and reverse key. Two locks formed by the hybridization of two sets of complementary strands, one set (red) placed on the lid (pink) and the other set (dark blue) on the plate (blue) initially keep the lid closed. The addition of key displaces the locks and thereby opens the lid. The addition of the reverse key displaces the key as a key-reverse key complex, leading to reclosing of the lock back to the initial state. **e–g** AFM images of cholesterol-free LGC without lid (**e**), with closed lid (**f**), and with opened lid (**g**). Scale bars, 50 nm. Source data are provided as a Source Data file.

Similarly, open-lid LGC-O appeared partly closed as the lid can obstruct the channel opening and the dimensions were 78.4 ± 3.5 nm and 21.8 ± 2.5 nm ($n = 13$) (Fig. 1g and Supplementary Fig. 6). The slightly larger-than-nominal dimensions of all LGC variants are likely due to the flattening of the negatively charged DNA plate on mica surface in presence of $Mg^{2+}$, compression by the AFM tip, or both factors[67]. The channels were lipid-tagged by incubating with cholesterol-modified DNA oligonucleotides that bound to designed sites at the bottom side of the large membrane cap (Fig. 1a and Supplementary Fig. 2).

**Nanopore interaction and insertion into lipid bilayers.** Following successful formation, we tested whether the cholesterol-modified channel can bind to and insert into lipid bilayers. To probe for membrane binding, channel variant LGC-N was added to small unilamellar vesicles (SUVs) and analyzed by agarose gel electrophoresis (AGE). The AGE tests were performed under conditions with and without $Mg^{2+}$, and we found that an $Mg^{2+}$ free buffer condition is essential to prevent non-specific adsorption to chelating lipid head-groups while still ensuring the stability of the LGC variants (Supplementary Note 1 and Supplementary Fig. 8). The stability of LGC in $Mg^{2+}$ free buffer was ascertained by AFM imaging (Supplementary Fig. 9). Increasing SUV concentrations led to concomitant gel electrophoretic upshift (Fig. 2a), suggesting that pores can bind efficiently to lipid membranes of slowly

migrating vesicles[68]. No gel-upshift was observed when LGC lacked cholesterol, underscoring its role for membrane binding. Cholesterol-mediated binding was also confirmed with direct visualization by TEM imaging (Fig. 2b, blue arrows and Supplementary Fig. 10). The extent of membrane binding was probed by incubating Cy3-labeled LGC-N with giant unilamellar vesicles (GUVs) and examination with fluorescence microscopy. Co-localization of the cholesterol labeled pore with the vesicle perimeter indicated successful membrane binding (Fig. 2c, d and Supplementary Fig. 11).

To explore whether LGC-N punctures lipid bilayers, we tested the influx of membrane-impermeable Atto633 dye into the interior of GUVs[48,50,51]. GUVs were placed in a solution of Atto633, and fluorescence microscopy tracked any changes in the fluorescence content of GUVs after adding LGC-N. The channel successfully inserted into, and punctured membranes as indicated by Atto633 signals that increased within GUVs over the full incubation time (Fig. 2d, top). A total of 59% of GUVs showed dye influx in case of LGC-N with cholesterol (LGC-N + Chol) (Fig. 2e and Supplementary Videos 2, 3), which compares to solely 3.5% of GUVs incubated with LGC-N lacking cholesterol (LGC-N-Chol) (Fig. 2e, bottom); membrane insertion via cholesterol is key for puncturing the bilayer for cargo transport (see Methods for details on how data were analyzed to identify influx). We further validated the membrane-spanning of our large

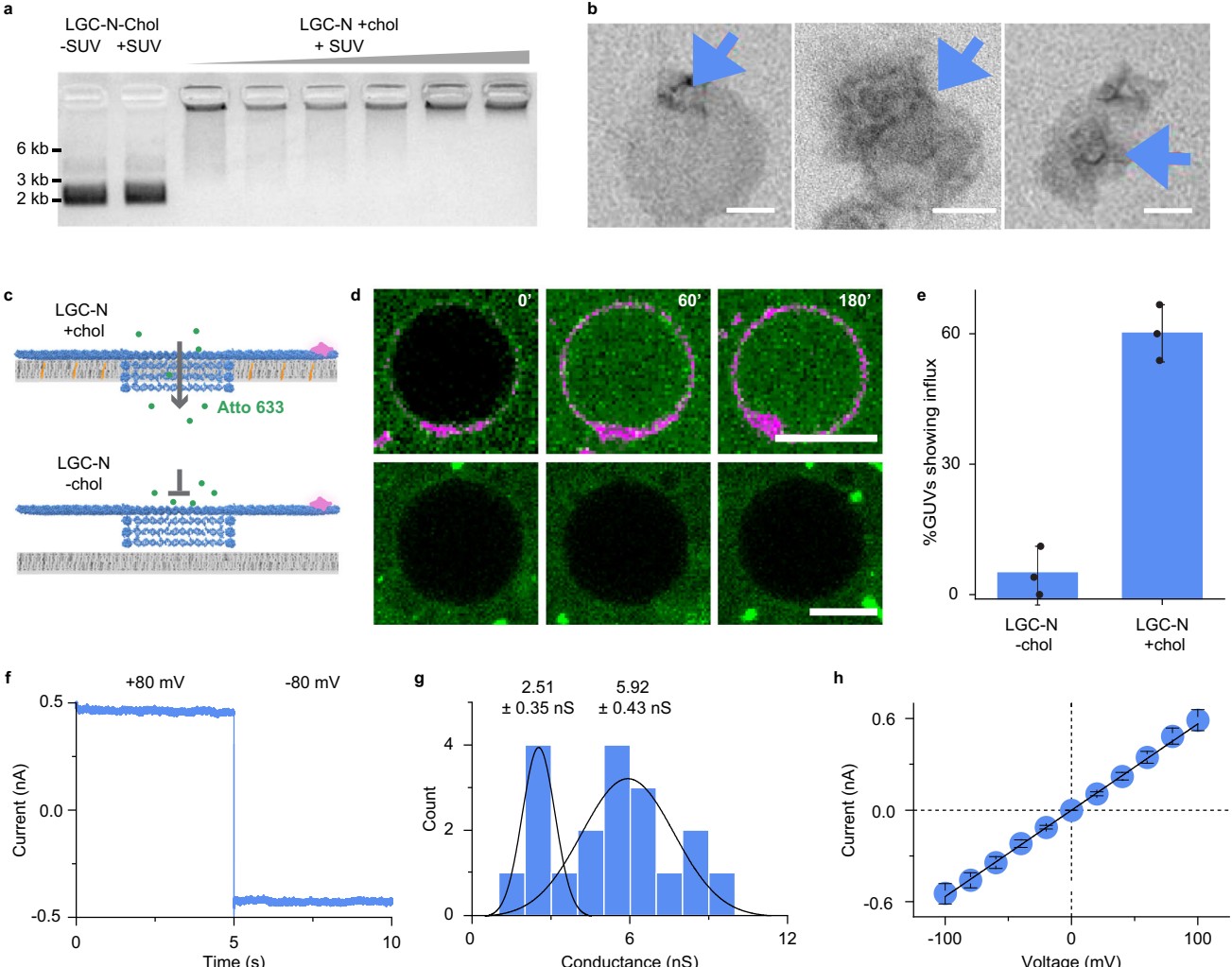

**Fig. 2 Insertion and interaction of LGC with lipid bilayers. a** 1.5% agarose gel analysis of the DNA pore and its binding activity to lipid membrane vesicles. Lanes from left to right: no lid LGC without cholesterol modifications (LGC-N-Chol) without SUVs, incubated with SUVs (DOPC/DOPE = 7:3, 0.5 mM total lipid), no lid LGC with 64 cholesterol modifications (LGC-N+Chol) incubated with SUVs (total lipid concentration 0, 0.01, 0.025, 0.05, 0.25, and 0.5 mM, respectively). Data are representative of more than three repeats. **b** Representative TEM images of LGC-N+Chol pores bound to lipid vesicles. Blue arrowheads pinpoint the pores. Scale bars: 50 nm. The data are representative of $n = 2$ independent experiments. **c** Scheme of the GUV-dye-influx assay and **d** a time series of corresponding confocal GUV images with Cy3-labeled LGC (magenta) and Atto633 dye (green). Top: LGC-N+Chol readily interacts with bilayer (magenta circle around GUVs). Their insertion leads to an influx of the Atto633 dye inside the GUV interior. Bottom: LGC-N-Chol does not interact with bilayer (no magenta circle around GUV) or insert into GUV, showing no dye influx over the course of 3 h. The data are representative of $n = 3$ independent experiments **e** Bar plot showing the percentage of GUVs with a filled interior after 3 h. The data summarize the average percentage of influx and error bars show the standard deviation of mean percentage influx counted from three independent experiments across $n = 49$ GUVs in case of LGC-N +Chol and $n = 59$ for LGC-N-Chol. Scale bars: 10 μm. **f** Current trace showing a single LGC-N+Chol channel inserted into a planar DPhPC membrane. The trace was recorded at an applied voltage of +80 mV for the first 5 s after which voltage potential was switched to −80 mV. **g** Conductance histogram of 19 individual LGC-N+Chol channels recorded at −20 mV. **h** Current-voltage (IV) plot showing the average current of 19 individual insertions ±SEM at membrane potentials ranging from −100 mV to +100 mV in 20 mV steps. All electrophysiological experiments were conducted in buffers composed of 1 M KCl, 10 mM HEPES pH 7.6. Source data are provided as a Source Data file.

and gated channel with a fluorescence resonance energy transfer (FRET) assay (Supplementary Fig. 12), which detects the interaction of a donor dye at the inserted channel with acceptor dyes in the GUV lumen.

The observed 59% influx via the LGC pores is significantly higher than the 8–11% increase detected in previous reports[43,48]. Moreover, the LGC mediated influx was completed within 1 h after channel addition, which is considerably faster than the 5–8 h completion time required in the case of previous DNA nanopores[43,48]. The differences in influx rates may be a result of several interdependent phenomena that lead to DNA nanopore-mediated transport: (i) nanopore binding to the membrane, (ii) reorientation of the stem

and puncturing of the lipid bilayer to form a channel, and (iii) the diffusion of the dye through the pore. The nanopore binding to the membrane is strongly governed by the number of cholesterol molecules, ionic concentration, and temperature. The reorientation is the rate-determining step and depends on the flexibility of cholesterol-bearing segments of the nanopore. The final step of dye diffusion is a function of the size of the dye molecules, medium viscosity, temperature, concentration gradient, and pore dimensions[43,48]. We attribute the faster flux in the case of LGC-N compared to previous pores due to the placement of a large number of cholesterol anchors to accelerate membrane binding (step i). The flexibility of the cholesterol-bearing one-layer origami plate allows

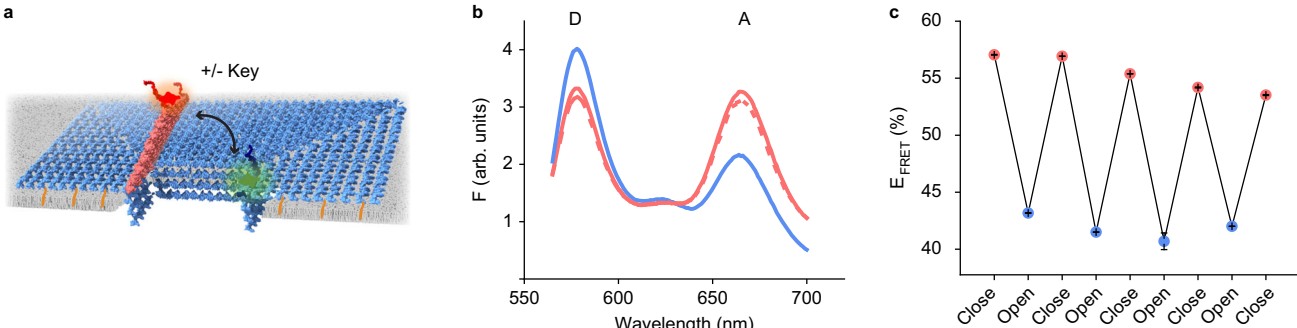

**Fig. 3 Reversible gating by LGC. a** Scheme showing LGC, containing donor dye on the plate (green) and acceptor dye on the lid (red). The addition of key and reverse key leads to respectively, opening and closing of the lid—resulting in closing or increasing the distance between the two dyes. **b** Representative fluorescence spectra for one full cycle of lid opening-closing (D and A on the graph represent fluorescence maxima for donor and acceptor respectively) and **c** FRET efficiency corresponding to reversible gating of LGC lid over four full cycles; data points and error bars respectively represent average relative FRET efficiency and standard deviation of the mean from $n = 3$ technical repeats. Initially, when the lid is closed, the spatial proximity of the donor and acceptor enables FRET, resulting in lower donor fluorescence, higher acceptor fluorescence (**b**, solid pink), and high FRET efficiency (**c**, close, pink dots). Opening the lid using a key (dynamic-open) moves the donor and acceptors far from each other. Thus, FRET ceases, resulting in higher donor fluorescence, lower acceptor fluorescence (**b**, solid blue) and lower FRET efficiency (**c**, open, blue dots). Closing the lid back from its open state using a reverse key (dynamic-closed) restores FRET, again resulting in lower donor fluorescence, higher acceptor fluorescence (**b**, dashed pink), and high FRET efficiency (**c**, closed, pink dots). Source data are provided as a Source Data file.

for rapid reorientation of the pores. Faster insertion kinetics were observed with LGC-N compared to previous origami pores that always placed the cholesterol molecules in a rigid framework, thereby slowing this reorientation step. Interestingly, the dye diffusion rate in LGC-N was approximately five times slower compared to the theoretically calculated time required for diffusion (Supplementary Fig. 13 and Supplementary Note 2). This is likely due to the large pore diameter slowing the rate-limiting reorientation step.

The membrane-spanning nature of LGC-N was confirmed by single-channel current recordings (Fig. 2f–h). Individual channels were inserted into a planar DPhPC lipid bilayer that separates two chambers filled with electrolyte (1 M KCl, 10 mM HEPES pH 7.6). To induce ion flow across an inserted channel, a transmembrane potential was applied. An ensuing steady current of 461 pA (Fig. 2f, +80 mV) indicated membrane insertion. LGC-N's average conductance values distribute into multiple conductance states (Fig. 2g and Supplementary Fig. 16), in line with previous measurements of large diameter DNA nanopores[49,51]. Two peaks are seen in the conductance histogram, a smaller peak with an average of $2.51 \pm 0.35$ nS ($n = 6$, ±error of Gaussian fit) and a larger peak with an average of $5.92 \pm 0.43$ nS ($n = 16$, ±error of Gaussian fit) (Fig. 2g). The linear relationship between current magnitude and voltage is expected for a channel lumen with vertical symmetry (Fig. 2h). Control experiments using fluorescence microscopy (Supplementary Fig. 14) and electrical current readout (Supplementary Fig. 15) of a flat origami plate without the stem established that the translocation of dye is indeed through the central pore and not through the toroidal pores known to form between the sides of DNA nanopores and the lipid membrane.

**Reversibly ligand-gated lid-controlled transport of small-molecule cargo.** Unlike constitutively open pores, biological ion channels usually open solely upon specific stimuli and then close by passive ligand dissociation. Attempts[48,53] have not been able to functionally replicate and transcend biology by actively controlling pore opening and closing via external triggers. Our large gated channel was designed to achieve defined transport control via a toehold-mediated strand displacement reaction[69,70] between lock and key (Fig. 1d and Supplementary Fig. 4). We tested the

reversible opening and closing of the lid using FRET. A Cy3 donor and Cy5 acceptor dye were placed in the lid and plate, respectively (Fig. 3a, D and A). Due to spatial proximity between the dyes, FRET occurred in the closed state (Closed or LGC-C) as reflected by the low donor fluorescence at $\lambda_{max}^{Cy3} = 564$ nm and high acceptor fluorescence at $\lambda_{max}^{Cy5} = 670$ nm (Fig. 3b, solid pink line). Opening the lid by adding a key (Dyn. Open) increased the distance between the reporter dyes and resulted in a low FRET, a higher donor emission, and a drop in the acceptor emission (Fig. 3b, blue line). Closing the lid back from its open state using a reverse key (Dyn. Closed) restored FRET, lowered donor fluorescence, and increased acceptor fluorescence (Fig. 3b, dashed pink line). The key-controlled switch was sequence-specific as confirmed by a mismatch key (Supplementary Fig. 17). Analysis of the kinetic FRET signal revealed that lid opening followed second-order kinetics[71] at a rate constant of $1940 \pm 50 \, M^{-1} \, s^{-1}$ until transport completion after 2 h (Supplementary Fig. 18 and Supplementary Note 3). Cyclic opening (Fig. 3c, blue dots) and closing (Fig. 3c, pink dots) of the lid was also demonstrated by comparing the FRET efficiency at each stage.

After establishing reversible gating, we utilized the GUV-dye influx assay to demonstrate controlled transport through the channel. GUVs were immersed in a solution of Atto633, and influx was probed via fluorescence microscopy. In control analysis, cholesterol-modified LGC-C bound to the vesicle membranes whereas the non-cholesterol version did not show binding, as indicated by the presence or absence of fluorescent rings around the GUV perimeter (Supplementary Fig. 19). Probing of the fluorescence intensity within GUVs established that membrane-bound LGC-C did not lead to dye flux into the vesicles (Fig. 4a, b, top; Supplementary Video 4). This implies that the lid completely blocked transport through the LGC channel. Upon addition of keys, however, dye fluxed inside GUVs demonstrating that the opened-lid channel is transport active (Fig. 4a, b middle and Supplementary Video 5). To confirm that the channel can be shut back to cease transport function, a "dynamically closed pore" was obtained by first, dynamically opening the closed LGC by addition of key, followed by addition of the reverse key. After dynamically closing the membrane-inserted LGC in the absence of Atto633, the dye was added to the GUVs. No dye influx was seen in this case (Fig. 4a, b, bottom and Supplementary Video 6), which

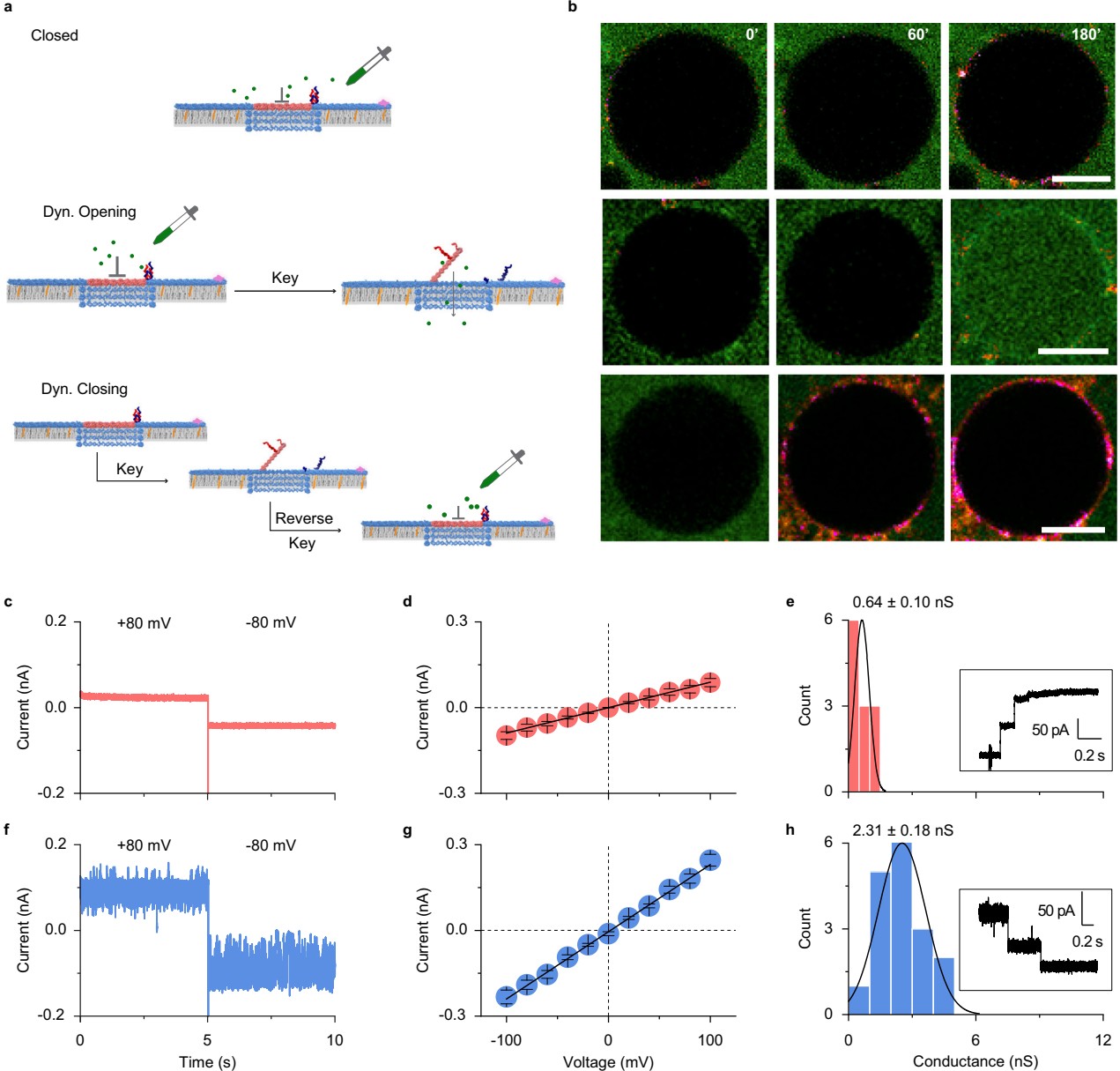

**Fig. 4 Ligand-gated LGC dynamically controls cargo transport across the lipid bilayer. a** GUV-dye-influx assay scheme. **b** A time series of corresponding confocal images of GUVs incubated with cholesterol and Cy3-labeled LGC (magenta) and Atto633 dye (green), dropper represents the time point at which Atto633 dye was added. Top—closed lid LGC does not allow Atto633 dye influx into GUV interior. Middle—dynamically opening the LGC lid with key results in the influx of Atto633 dye into GUV. Bottom—separate experiment in which adding Atto633 to dynamically closed LGC does not lead to any dye influx into GUVs, showing dynamic closing is effective. Scale bars: 10 μm. The data were representative of $n = 3$ independent experiments. **c** Current trace of a single LGC-C channel inserted into a planar DPhPC membrane. The trace was recorded at a voltage of +80 mV for the first 5 s, after which the potential was is switched to −80 mV. **d** Current-voltage (IV) plot showing average current of 12 individual insertions ± standard error of the mean of LGC-C at membrane potentials ranging from −100 to +100 mV in 20 mV steps. **e** Conductance histogram of 12 individual insertions of LGC-C obtained at −20 mV. Inset shows an example trace on the transition from closed-lid pore to open-lid after addition of 15 nM key. **f** Current trace of a single LGC-N + Chol channel inserted into a planar DPhPC membrane. The trace was recorded at a voltage of +80 mV for the first 5 s, after which the voltage potential was switched to −80 mV. **g** Current-voltage (IV) trace showing average current of 17 individual insertions ± standard error of mean of LGC-O at membrane potentials ranging from −100 to +100 mV in 20 mV steps. **h** Conductance histogram of 19 individual LGC-N + Chol channels recorded at −20 mV. Inset shows an example trace on the transition from open-lid pore to closed-lid after addition of 15 nM reverse key. All electrophysiological experiments were conducted in buffers composed of 1 M KCl, 10 mM HEPES pH 7.6. Source data are provided as a Source Data file.

confirms that the dynamically closed LGC blocks transport. Negative control experiments with LGC-C without cholesterol modification (Supplementary Video 7) and the mismatch keys (Supplementary Fig. 20) did not show similar Atto633 dye influx. Moreover, a two-color confocal microscopy experiment was performed (Supplementary Fig. 21). Firstly, red dye (Atto633)

was allowed to influx into the vesicle through the open lid pore LGC-O. Closure of the lid did not allow a second green dye (Atto 488) to the influx, demonstrating successful closure of the opened lid. These data clearly confirm that the lid can be reversibly opened and closed with external triggers. Although previously one-way or partial opening[53] and closing[43] has been shown in

separate reports, dynamic and reversible gating has not been shown in a synthetic nanopore before.

Single-channel current recordings probed the characteristics of cholesterol-modified closed lid (LGC-C) and open-lid (LGO-O) LGCs (Fig. 4c–h), and the dynamic transitions between the two. As would be expected from the steric blockade by the closed lid, LGC-C featured a far smaller current and corresponding conductance at $0.64 \pm 0.10$ nS ($n = 12$, ±error of Gaussian fit) (Fig. 4c, e) than had been obtained with no lid LGC at $5.92 \pm 0.43$ nS ($n = 15$, ±error of Gaussian fit) (Fig. 2f, g). The presence of the lid did not affect linear voltage-current dependence (Fig. 4d). The small residual current of LGC-C might stem from ion leakage either through the periphery of the pore (at the DNA duplex/lipid interface)[51–53] or through the DNA duplexes of the lid, or a combination of both. By comparison, the conductance of LGO-O at $2.31 \pm 0.18$ nS ($n = 17$, ±error of Gaussian fit) (Fig. 4f, h) is almost four times the value of LGC-C (Fig. 4e), reflecting that the open lid allows more ion transport. However, it is still more than half the value of no-lid LGC in agreement with a residual blockade by an open lid (Fig. 2f, g). Control experiments were performed using a DNA plate construct without a pore-forming cap region (Supplementary Fig. 15). These constructs did not produce pore-forming currents, highlighting the importance of the cap region for lipid bilayer penetration.

The single-channel analysis provided further insight into the dynamic nature of the LGC lid. Current traces of the single LGC-O channels had more high-frequency fluctuations (Fig. 4f and Supplementary Figs. 22b, 23) than the less noisy trace of LGC-C (Fig. 4c and Supplementary Figs. 22a, 23) at all voltages from −100 to +100 mV (Supplementary Fig. 22). The current fluctuations likely reflect the dynamic movements of the lid to and from the channel base in LGC-O compared to the static lid in LGC-C. Indeed, electrophoretically driving the lid of LGC-O to its base plate via a negative potential led to an additional current level at $-37.8 \pm 15.6$ pA ($-60$ mV, ±STD) next to the main conductance peak at $-96.7 \pm 7.8$ pA ($-60$ mV, ±STD) (Supplementary Fig. 22), an increase in noise at negative potentials shown by power spectrum noise analysis (Supplementary Fig. 24), and an increase in event frequency at negative potentials (Supplementary Fig. 25). By contrast, moving the lid away from the plate via positive potentials only led to a single, main peak at $78.5 \pm 5.8$ pA (60 mV, ±STD) (Supplementary Fig. 22) and resulted in less noise (Supplementary Fig. 24) and fewer blocking events (Supplementary Fig. 25). The voltage-dependent noise strikingly reveals nanomechanical-dynamic changes of a DNA structure at the single-molecule level.

Single-channel analysis revealed insight into the closing and opening mechanism. LGC-O could be closed by adding a reverse key, as demonstrated by the transition from an open state current at $97.4 \pm 5.2$ pA (10 mV, ±STD) to the lower-amplitude closed state at $17.7 \pm 2.9$ pA (10 mV, ±STD) (Fig. 4h, inset and Supplementary Fig. 26). This transition occurred in a stepwise fashion, likely because the two lid locks are closed one after the other leading to an intermediate state of one closed and one open lock (Fig. 1d). Similarly, LGC-C was opened by adding key as indicated by a switch from the closed state at an amplitude of $8.7 \pm 2.7$ pA ($-10$mV, ±STD) to the open-state at amplitude $146.9 \pm 2.8$ pA ($-10$ mV, ±STD) (Fig. 4e, inset and Supplementary Fig. 26). This transition occurred also in a stepwise fashion. Our data reveal the opening and closing mechanism of the LGC channel in unprecedented detail and indicates ways to fine-tune the opening by varying the lock number.

**Transport of folded proteins across the nanopore**. We finally exemplified the power of large and gated channels by regulating the flux of folded proteins across membranes via defined lid

opening and closing. GUVs were immersed in a solution of green fluorescent protein (GFP, hydrodynamic diameter = 5.6 nm[72]). Cholesterol-modified channels bound to GUV membranes whereas unmodified versions did not bind (Supplementary Figs. 27, 28 and Supplementary Videos 8, 12). When influx was monitored by determining fluorescence intensity in GUVs, control channel LGC-N with cholesterol without lid led to transport across membranes (Supplementary Fig. 27 and Supplementary Video 8). By contrast, cholesterol-modified closed-lid LGC-C blocked protein flux (Fig. 5a–c, top and Supplementary Video 9). However, adding the key opened up the transport function (Fig. 5a–c, middle and Supplementary Video 10). Again, in the dynamically closed pore, no protein influx was observed (Fig. 5a–c, bottom and Supplementary Video 11). Negative control experiments with 500 kDa FITC-dextran featuring a hydrodynamic diameter of 31.8 nm[73] established that channel transport did not occur due to size-exclusion (Supplementary Fig. 29); FITC-dextran also ruled out membrane rupturing. Another control experiment using pre-encapsulated cargo in GUVs demonstrated that LGC can release nucleic acid cargo such as dye-labeled DNA (Supplementary Fig. 30) which can prove to be valuable for vaccine delivery. The successful data on the precisely timed transport of folded proteins and small organic dyes with equal ease (Fig. 4a, b and Supplementary Fig. 31) is beyond the scope of biological and any previously engineered membrane channels.

The transport of folded protein across the membrane-nanopore lumen was examined with single-channel current recordings (Fig. 5d–g). As a model protein, trypsin (hydrodynamic diameter of 4.0 nm[74]) and net positive charge (pI 10.1, pH 7.6) was used. No translocation events occurred upon the addition of 6.6 μM trypsin to the *cis* chamber of membrane inserted LGC-C (Supplementary Fig. 32) as shown by relatively steady current flow (Supplementary Fig. 32a, b). By contrast, with LGC-N channels increasing concentrations of trypsin led to blockades (Fig. 5d–g and Supplementary Fig. 33), which clustered into two event types (Fig. 5d–g). When analyzed by their relative percentage blocking amplitude, $A/I_O$ (amplitude of block/ open current amplitude), and dwell time, $\tau_{off}$, (Fig. 5f, g), events clustered at $1.88 \pm 1.5\%$ (Type I) and at $10.8 \pm 4.8\%$ $A$ (Type II). We suggest that Type I events occur due to the brief interaction of the protein with the nanopore at the lumen opening (Fig. 5d), while in Type II events proteins fully translocate through the nanopore (Fig. 5e). In order for translocation events to be detected, intermittent interactions of the positively charged trypsin to the negatively charged pore wall must occur[50,75]. Voltage-dependent analysis (Supplementary Fig. 34) revealed that Type II blocking events have longer dwell times at lower voltages ($1.65 \pm 0.10$ ms at $-20$ mV) than higher voltages ($0.28 \pm 0.04$ ms at $-50$ mV) in line with the expected faster translocation of trypsin through the pore at higher voltages.

The transport of folded GFP through LGC-N was also studied. GFP is larger than trypsin (hydrodynamic diameter of 5.6 nm[72]) and has a lower isoelectric point (pI 5.8) and will therefore have a net negative charge in standard electrophysiological buffers of pH 7.6. Little interaction of GFP with the nanopore is expected at neutral pH as intermittent electrostatic binding of a positively charged protein with the negatively charged DNA pore wall is thought to be necessary to resolve translocation events[50,75]. Indeed, when 310 nM GFP was added at neutral pH to LGC-N no obvious translocation events could be seen (Supplementary Fig. 35a–c). However, when the pH of the electrophysiological buffer was dropped below the pI of GFP to pH 4.5 clear blockade events could be detected (Supplementary Fig. 35d–f) suggesting electrostatic interaction between GFP and the pore wall. GFP blockade events clustered into two types of events when each

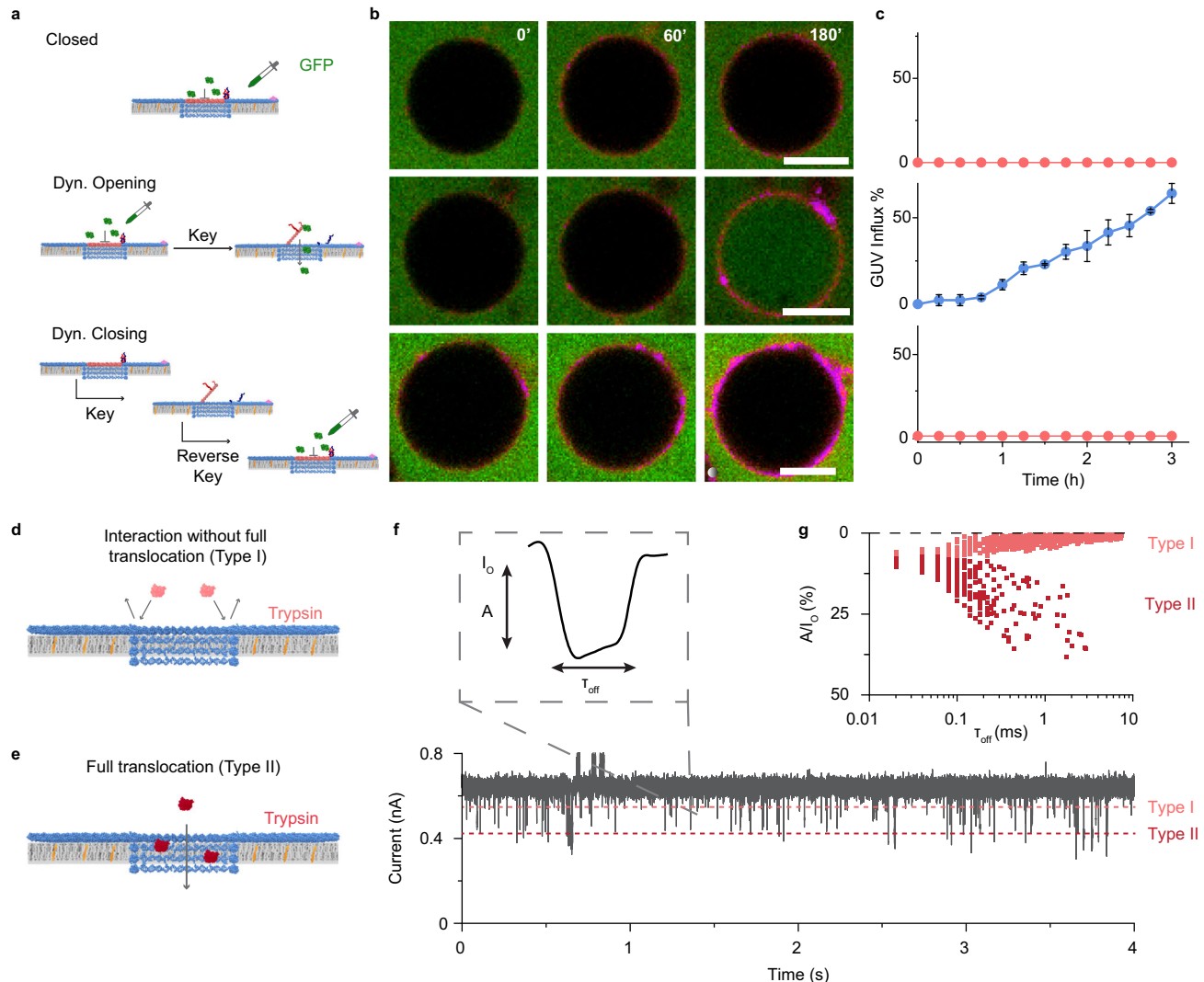

**Fig. 5 Reversibly gated protein transport by large diameter nanopore. a** Scheme of GFP-influx assay using GUVs. **b** Respective time series of confocal images at the given time points showing Cy5-labeled LGC (magenta) and GFP (green). Scale bars: 10 μm. **c** Percentage of GUVs showing influx over time for Closed, Dyn. Open, and Dyn. Closed versions of LGC. The plots show the average percentage of influx and standard deviation from $n = 125$, 255, and 125 GUVs, respectively, from three independent experiments. See Methods section on how data were analyzed to identify influx. **d, e** Scheme depicting the electrophysiological characterization of trypsin transport through an LGC showing Type I (**d**) and Type II (**e**) events. **f** Representative single-channel current trace of LGC-N at +80 mV after the addition of 6.6 μM trypsin to the *cis* chamber obtained at 50 kHz sampling rate. Inset shows an isolated translocation event characterized by dwell time ($\tau_{off}$) and blocking amplitude ($A/I_O$). **g** A scatter plot of individual trypsin translocation events based on $\tau_{off}$ and $A/I_O$. Electrophysiological experiments were conducted in buffers composed of 1 M KCl, 10 mM HEPES pH 7.6. Source data are provided as a Source Data file.

event was plotted in relation to its blocking amplitude and dwell time (Supplementary Fig. 36), similar to the blockades caused by trypsin. Type I GFP events clustered at 3.8 ± 3.7% and Type II events clustered at 26.1 ± 7.3%. The latter events are interpreted to fully translocate through the nanopore. The bigger blocking amplitude of GFP is expected for the protein's larger size (hydrodynamic diameter = 5.6 nm) compared to trypsin (hydrodynamic diameter = 4.0 nm), in line with the previous reports[50].

**Advantages and limitations of pore design**. In this work, we set out to construct a reversibly gated DNA nanopore with a luminal diameter large enough to accommodate proteins for transport assays. Whilst achieving these goals, characterization of the pore has highlighted some key advantages of the DNA nanopore design over existing artificial nanopore designs, as well as revealing some limitations of the design. Discussing the

advantages and limitations here could help future researchers improve the design and sensing abilities of DNA nanopores, accelerating the advancement of research in the field of DNA nanostructures.

One advantage of the pore design in this study lies in the nanomechanical control of pore activity, which enables controlled transmembrane transport of an array of cargoes ranging from small-molecule dyes to large folded protein. Fluorophore flux studies have shown high fidelity of dye transport through open lid nanopores, with no such transport seen in closed lid nanopores (Fig. 4a, b). However, single-channel current recordings revealed some inherent ionic leak through the closed lid nanopore (Fig. 4c–f), with an average conductance of 0.64 ± 0.10 nS, highlighting the high sensitivity of these single-molecule experiments. Such leaks have been shown previously with a closed lid DNA nanopore[53]. Leak currents likely stem from toroidal pore formation of lipid bilayer spanning DNA duplexes[51,52], as well as

ionic flow through the nanopore lid[76] (Supplementary Fig. 37). Dye transport studies (Fig. 4a, b) show that the closed lid nanopore is not permeable to the larger dyes (Atto633 diameter = 1.3 nm), suggesting the leak of ions only (chloride ion Stoke's radius = 0.1 nm). Major groove binders[77] or hydrophobic coating[78] have been proposed as potential strategies to reduce leak currents of DNA nanostructures.

In this study, open lid nanopores have been successfully designed to controllably accommodate the folded proteins trypsin and GFP for fluorophore release flux studies (Fig. 5a–c) and single-molecule electrophysiological transport studies (Fig. 5d–g) which is a major advantage over current small-diameter protein-based nanopores. However, single-channel current recordings have revealed smaller experimental conductances of no lid nanopores when compared to designed pore diameters. No lid nanopores have an average conductance of 5.92 ± 0.43 nS, with protein transport studies of both trypsin and GFP attributing average current blocks of 11 and 26% respectively. Taking into consideration these measurements, it is therefore likely that the pore does not have the designed pore size of 20 nm, but is closer to 5–8 nm when measured in a planar lipid bilayer (based on estimated protein volumes). Whilst there is an apparent reduction in pore diameter, this is still of a size large enough to accommodate folded protein. Previous research has shown buckling and voltage-dependent conformational changes of DNA origami structures[79,80] and compression of lipid spanning helices of the lumen[50] that may account for low experimental conductances. Electro-osmotic effects due to the charged DNA backbone[50,81] as well as increased access resistance in nanopores with low thickness-to-diameter ratios and charged surfaces[82] may also have integral roles in reducing experimental conductance. Future molecular dynamic studies[52,76,83] could help explain the conductance discrepancies in large diameter DNA nanostructures.

Another advantage of our nanopore design is the horizontal routing approach that allows high design flexibility, which means making small adjustments to tailor the pore to a required function is not a labor-intensive process. For example, tailoring pore diameters for size-dependent analysis of folded proteins, or adjusting lid opening/closing for real-time analysis of protein folding interactions. Our protein transport studies in planar lipid membranes have shown that relatively high concentrations of trypsin (between 66 nM and 6.6 μM) are required for adequate analysis of translocation events (Supplementary Fig. 33). Protein translocation under voltage is a rapid process, with the majority of proteins passing through the nanopore with dwell times <20 μs, too fast to detect with electrophysiological equipment. Therefore, protein translocation can only be detected in this case when trypsin translocation is slowed, such as via interaction with the negatively charged DNA of the pore lumen. This interaction has been noted previously with trypsin translocation through a DNA nanopore[49], and is in concordance with simulation data of trypsin translocation through a DNA nanopore[75]. Only trypsin that has adequate interaction with the DNA pore lumen will therefore be detected, accounting for the larger concentration of protein required in these translocation experiments. Future work could look at adding protein-specific tags inside the pore lumen to increase the specificity of protein interactions, whilst also increasing protein dwell times.

## Discussion

Here we pioneered a large diameter DNA nanopore with a sequence-specific, fully reversible gating. By controlling the transport of various cargoes ranging from small-molecule dye to folded proteins, the DNA pores exceed the natural analog—ligand-gated ion channels which have much narrower pore dimension. The DNA nanopore is also innovative as it offers reversible and tunable pore gating to control transmembrane transport of protein cargo. The design freedom offered in the current work may enable the creation of pores to reversibly capture proteins and study their interaction with free-flowing ligands[84,85] or real-time protein folding and unfolding[86]. The reversibly controlled flow of cargo by a nanomechanical lid may also be exploited to transport bioactive cargo into cells, or to construct cell–cell communication[61] for artificial gap junctions and to enable integration of artificial tissues with living cells and tissues for tissue engineering[62]. In conclusion, by offering solutions to several challenges in nanopore sensing our study extends the versatility and scope of artificial nanopores beyond nature and thus opens various exciting applications.

## Methods

**Materials**. All DNA oligonucleotides were purchased from Integrated DNA Technologies Inc. with standard desalting unless mentioned otherwise. The strands used for dynamic reconfiguration were purified in-house with denaturing PAGE gels. Lipids were purchased from Avanti Polar Lipids Inc. All other chemicals (e.g., sucrose, glucose, FITC-dextran) were purchased from Sigma Aldrich Inc.

**Large gated channel design and assembly**. The nanostructures were designed de novo by Tiamat software[65]. Information on the DNA oligonucleotide sequences, two-dimensional DNA maps are provided in Supplementary Data 1 and Supplementary Fig. 2. Structures were assembled by mixing a final concentration of the scaffold strand at 50 nM and staples at 5x excess concentration in 12.5 mM MgCl$_2$ in 1x TAE buffer (40 mM Triacetate and 1 mM EDTA, pH 8.3). The structures were folded in Life Technologies SimpliAmp thermal cycler using the following annealing protocol—10 min consecutively at 90, 80, 70, and 60 °C; 20 min consecutively at 50, 40, 30, and 20 °C. Following annealing, the structures were stored at 4 °C until further use.

**Purification of DNA nanopores**. The folded DNA origami nanopores were purified either by filter centrifuge or by agarose gel purification (see the native agarose gel electrophoresis section below). For a typical purification using Amicon centrifugal filter (100 kDa MWCO), the filter was firstly washed with 1× TAE-Mg buffer (20 mM Tris base, 10 mM acetic acid, 0.5 mM EDTA, 12.5 mM Mg(OAc)$_2$, pH 8.3), the origami solution was then added and spun at 1500 × $g$ for 2 min, and followed with five additional spinning steps, washing with 400 μl 1× TAE-Mg buffer before each step. The purified samples were stored at 4 °C until further use.

**Native agarose gel electrophoresis**. For characterizing the formation of the pore, a 1.5% agarose gel was casted 1× TAE-Mg buffer. The gel was run at 120 V in an ice-water bath for 1.5 h. Sybr green stain was added with the sample for gel imaging. For gel purification, after gel electrophoresis, the gel was illuminated under a UV lamp (365 nm) and bands of interest were carefully cut out from the gel. The gel pieces were frozen at −20 °C and the purified nanopores were recovered using Freeze 'N Squeeze spin columns. To make Mg$^{2+}$ free nanopore samples, the solutions were buffer exchanged using an Amicon filter. The method was the same as that for filter centrifuge purification above except that the buffer used here was Mg$^{2+}$ free: 50 mM HEPES (pH 7.6) supplemented with 500 mM NaCl. The purified pores, optionally incubated with SUVs, were analyzed using 1.5% agarose gel electrophoresis in 0.5 × TAE buffer (20 mM Tris base, 10 mM acetic acid, 0.5 mM EDTA, pH 8.3) and 0.5 ×TAE-Mg buffer (20 mM Tris base, 10 mM acetic acid, 0.5 mM EDTA, 10 mM MgCl$_2$, pH 8.3) at 65 V in ice-water bath for 1.5 h. Uncropped and unprocessed scans of all the gels are provided in the source data.

**Atomic force microscopy (AFM)**. Annealed sample (3 μl) was deposited onto a freshly cleaved mica surface (Ted Pella) and 1× TAE-Mg$^{2+}$ buffer was added immediately to the sample. After about 30 s, 3 μl NiCl$_2$ (25 mM) was added. An extra 60 μl of the same buffer was deposited on the AFM tip. AFM imaging was performed in the "ScanAsyst mode in fluid" on the Dimension FastScan, Bruker with the Scanasyst-Fluid+ tips from Bruker.

**Transmission electron microscopy (TEM)**. Structures (10 μl) were added to plasma-treated TEM grids (Agar Scientific, AGG2050C) for 1 min or negative stain carbon B type grid for 10 min and wicked off. The sample was stained for 10 s with 2% uranyl formate or 1% uranyl acetate with 2 mM NaOH. The stain buffer was then wicked off and the grid air-dried for 20 min. The grid was then imaged using a JEM-2100 electron microscope (equipped with an Orius SC200 camera) or Philips TM 12 TEM operated at 120 kV at 33000× to 80000× magnification.

**Preparation of small unilamellar vesicles (SUVs)**. A lipid solution (DOPC/DOPE = 7:3, 10 mg/mL in chloroform) dispensed in a 2-mL-glass vial was blown dry with argon airflow. The dried lipid film was then suspended in 50 mM HEPES (pH 7.6) supplemented with 500 mM NaCl, and treated by sonication for 30 min. the solution was then put through 10 freeze-thaw cycles in liquid nitrogen and 60 °C water respectively.

**Cholesterol modification of nanopore**. Cholesterol-modified strands were purchased from Integrated DNA Technologies Inc. with PAGE purification. The strand was dissolved in water to make a 100 μM solution. Immediately the solution was made into 5, 10, and 20 μl aliquots and lyophilized. This was done in order to avoid aggregation of the cholesterol-modified strands in water. The structures were then incubated with 2× excess concentration of cholesterol-modified strands (64 cholesterol-modified positions on the structure were accounted for) at 37 °C for 12 h.

**Preparation of giant unilamellar vesicles (GUVs) for confocal measurements**. The GUVs were prepared by the inverted emulsion method[50]. POPC (150 μl, 10 mM) in chloroform was added to a 1 mL glass vial, the solvent was removed under vacuum and rotation using a Buchi rotary evaporator set at high vacuum for at least 30 min. The thin film generated was resuspended in mineral oil (150 μl) by vortexing and sonicating for 10 min. About 25 μl of inner solution (IS, the solution that will be encapsulated inside GUVs) containing ~435 mOsm/kg sucrose was added to the mineral oil. A water-in-oil emulsion was created by suspending the IS into the mineral oil by pipetting up and down approximately ten times followed by vortexing at the highest speed for ~30 s and sonicating for 10 min at room temperature. This emulsion was then carefully added to the top of a 1 mL external solution (ES, the solution to be kept outside the GUVs) containing ~435 mOsm/kg glucose in a plastic microcentrifuge tube. The osmolarities of the IS and ES were measured by an osmometer (Advanced Instruments Model 3320 Osmometer 2996) and balanced properly such that the osmolarity difference is less than 20 mOsm/kg. The GUVs were generated by centrifuging at 21,000 × g at 4 °C for 15 min. The mineral oil top layer and most of the sucrose layer (~900 μl) was carefully removed by pipettor, every time using a fresh tip. The remaining solution containing the pelleted vesicles was gently mixed with a pipettor, then transferred to a clean plastic vial leaving a small quantity to avoid contamination of the remaining trace amount of mineral oil to the GUVs.

**Confocal dye/protein influx assay**. GUV solution (3 μl) was mixed with 20 nM of cy3/cy5-labeled-nanopore and 2 μM of dye (Atto633) or protein (GFP). The total solution was made up to 30 μl by maintaining the osmolality balance of the inside and outside of the GUVs using a buffer containing 1 M HEPES and 100 mM NaCl. This solution was added to an ibidi μ-Slide 18 Well–flat slide. The slide was then centrifuged at 1000×g for 10 min to make sure that the GUVs are settled at the bottom of the slide. Then GUVs were then imaged on Nikon C2 Laser Scanning Confocal microscope at 40× magnification, 1.3 numerical aperture (NA) using a humidity chamber maintained at 32 °C (dye)/37 °C (GFP). Images were taken over 3 h at multiple points. For the GUV-influx studies with the dynamically closed LGCs, we could not stabilize the GUVs for more than 6 h under the imaging conditions as required for these experiments. Bursting of GUVs due to osmolarity imbalance caused by addition of reverse key prevented further measurements. Thus we performed a separate experiment by dynamically closing the channels in GUV membranes and then adding dye or GFP. The reversible gating of the pore is still functional as shown by the FRET as well as dye influx experiments.

**Calculating percentage influx**. The mean intensity inside each GUV in a frame was measured using ImageJ software. For each experiment, the intensity inside the GUVs at $t = 0$ min ($I_0$) and at $t = 180$ min ($I_{180}$) was measured and normalized by the average intensity of the background at the same time point as $I_{t,norm} = \frac{I_{t,i}}{I_{t,b}}$; where: $I_{t,norm}$ = normalized GUV interior intensity at time t; $I_{t,i}$ = GUV interior intensity at time t and $I_{t,b}$ = average background intensity at time $t$. GUVs that satisfied the threshold criteria ($I_{180,norm} - I_{0,norm}$) > 0.5 were considered to have shown influx over 3 h and from this the percentage of GUVs showing influx was calculated.

**Single-channel current recordings**. For planar lipid bilayer electrophysiological current measurements, integrated chip-based, parallel bilayer recording setups (Orbit Mini; Nanion Technologies, Munich, Germany) with multielectrode- cavity-array (MECA) chips (IONERA, Freiburg, Germany) were used[68] unless stated otherwise. Bilayers were formed of DPhPC lipid dissolved in octane (10 mg/mL). The electrolyte solution was 1 M KCl and 10 mM HEPES, pH 7.6. For pore insertion, a 2:1 DNA nanopore and 0.5% OPOE (n-octyloligooxyethylene, in 1 M KCl, 10 mM HEPES, pH 7.6) was added to the cis chamber; the trans side was electrically grounded. Successful incorporation was observed by detecting current steps. Current traces were acquired at 10 or 50 kHz where specified and subsequently Bessel-filtered, using Element Data Recorder software (Element s.r.l., Italy).

Single-channel analysis was performed using Clampfit (Molecular Devices, Sunnyvale, CA, USA).

**LGC scheme creation and simulation**. The LGC schemes were generated by converting the original Tiamat structures to.pdb structures using the TacoxDNA webserver[87] and UCSF chimera molecular visualization tool[88]. The simulations shown in Supplementary Fig. 6b were generated using oxDNA simulations[66] in oxDNA.org.

**Reporting summary**. Further information on research design is available in the Nature Research Reporting Summary linked to this article.

## Data availability
The authors declare that the source data supporting the findings of this study are available within the paper and its Supplementary information files. Source data are provided with this paper.

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

## Acknowledgements

This research was funded by the National Science Foundation (1644745), EPSRC (EP/N009282/1), the BBSRC (BB/M025373/1 and BB/N017331/1), and the Wellcome Institutional Strategic Support Fund and Moorfields BRC. The authors thank Dr. Honor Glenn of the Advanced Light Microscopy Core, Arizona State University's Biodesign Institute for consultation regarding confocal microscopy. We thank Dr. Rizal Hariadi for his generous contribution in purchasing the osmometer, critical for the GUV experiments. We thank Dr. Yan Liu, Dr. Xiang Lan, and Erik Poppleton for crucial discussions regarding data analysis, experiment designs and Dr. Jonathan R. Burns for his crucial input in preparing the figures.

## Author contributions

S.H. and H.Y. conceived the idea of the project. S.D. and F.Z. designed the nanopores and performed AFM experiments for the formation of the nanopore. S.D and L.A. designed and performed the FRET experiments for dynamic opening and closing of the nanopore. S.D and L.A. designed and performed experiments and data analysis for confocal microscopy images for the GUV influx assays. Y.X. and S.D. performed TEM imaging of the nanopore formation. Y.X. and S.D. performed TEM imaging of the nanopore interaction with SUVs. Y.X. carried out agarose gel experiments for studying the interaction of nanopores with SUVs. Y.X. prepared nanopore samples for the electrophysiological measurements. A.D. carried out the electrophysiological characterization of the nanopore including analysis and interpretation. I.Z. helped S.D. and L.A. to optimize GUV formation protocol. S.D. wrote the manuscript with inputs from L.A., A.D., Y.X., and F.Z. under the supervision of S.H. and H.Y.

## Competing interests

The authors declare no competing interests.
