## [Peer Review File · Nature Communications]

REVIEWER COMMENTS

Reviewer #1 (Remarks to the Author):

The manuscript by Dey et al. describes a membrane-spanning DNA pore with the largest diameter to date. With a 20nm-wide inner opening, the channel itself is about twice as large as the structurally more stable pore by Thompson et al (Nat. Commun., 2019, ref. 50) with an inner diameter of 9nm. This and other previous pores had the capability to open up in a stimuli-responsive manner by strand displacement. The advance here is that not just opening but also subsequent closure was achieved, although the process is not repeatable. As an additional aspect of novelty, the authors showed transport of a 32kDa protein across the membrane. It should be noted, however, that 40kDa Dextran was transported through DNA nanopores before (Nat. Commun., 2019, ref. 50). In principle, the level of novelty justifies publication – I leave it up the editor on what level – but the data and the presentation of the results is to large extends questionable. In parts it was difficult to evaluate the manuscript in a fair way because statistical information is missing. Several key aspects have to be addressed thoroughly.

Comments:

- 1) In several instances, one gets the impression that the authors are not quite open about what has already been done to increase the apparent novelty of the current work – partially disregarding their own work. It should be explicitly mentioned that they and others have already demonstrated pore opening by strand displacement and that cargo of similar molecular weight as their protein was already transported through DNA pores.
- 2) The authors state that their pore is 416 nm²-wide. This is misleading, since width is, per definition, not an area. Width is defined as “the measurement or extent of something from side to side; the lesser of two or the least of three dimensions of a body”. So width should be measured in nm and not nm².
- 3) The authors currently highlight the reversibility of the pore opening, but do not discuss at all that they currently can only open the pore once and then close it once. This is not a problem, but they should be open about it and maybe suggest strategies to achieve true reversibility.
- 4) Minor comment: From the main it is not so obvious with how many cholesterol-moieties the pore was functionalized. This should be stated in the text and the figure caption so that the reader does not have to search in the SI.

5) There is little evidence for the correct formation of the stem part of the pore and there are actually no cholesterol-moieties positioned on the stem. I wonder if it is structurally stable enough to penetrate the membrane when the cholesterol inserts. It is unclear how long the linkers for the chol-DNA are. Removal of a few strands on the stem would allow it to collapse completely. The current data seems to suggest some disturbance of the membrane, but with this concern it has to be interpreted even more carefully.

6) Minor comment: "To probe for membrane binding, channel variant LGC-N was added to small unilamellar vesicles (SUVs), and analyzed by gel electrophoresis in Mg²⁺ free buffer to avoid non-specific adsorption to chelating lipid head-groups". Have the authors established that the pore is even stable in Mg²⁺-free buffer? This still has to be done.

7) Minor comment: Figure 2c: How large is the scale bar?

8) Figure 2d shows a bar plot of the percentage of GUVs showing a filled interior after 3 hours. It is unclear how the data was analyzed and how the authors distinguished a vesicle with a filled interior from an unfilled interior. It would be helpful to include a plot with more information, namely to plot the Intensity inside the vesicle over the Intensity of the surrounding. What do the authors mean by percentage of influx? How many independent experiments were conducted?

9) The authors explicitly highlight the faster influx kinetics compared to the previous publication of Thompson et al. but they never show the influx over time (even though this should be straightforward since they monitor the process with fluorescence microscopy). They explain the faster kinetics as follows: "The faster flux is likely due to the larger pore lumen of LGC, the higher number of our inserted channels, or a combination of both." Have they checked the plausibility of this statement? My guess is that for a pore of that size, one would expect much faster equilibration once it is inserted. The authors should do some back of the envelope calculations and show the individual influx traces.

10) I am very surprised how messy the GUV preparation looks. I could well imagine that this is due to the inverted emulsion method. Is it possible that the authors did not control the humidity in the lab? Since encapsulation is not needed in this case, it would be much preferred to use the electroformation method. This avoids oil traces and should give much cleaner results.

11) I am somewhat surprised that the authors claim that Atto633 cannot diffuse through the closed pore. A single layer of DNA is porous itself and I would expect that a small dye molecule may well be able to penetrate through. Additionally, it is known that DNA pores form a toroidal pore at the DNA-lipid interface which should transport small dye molecules – especially for this pore which has no hydrophobic entities on the stem. Can the authors discuss why this should not be the case? It would be ideal to prove it with an independent experiment. Have the authors looked at longer time scales? Especially since the residual current of the closed pore is still half of the current of the open pore, I would not expect complete blockade of the fluorophore transport.

12) "These data clearly confirm that the lid can be dynamically opened and closed specially with external triggers, something which has not been achieved before by any means." Since opening has been achieved, this is a misleading statement.

13) “The small residual current of LGC-C might stem from ion leakage either through the periphery of the lid or through the DNA duplexes of the lid, or both.” The small residual current is not small at all, it is half of the open current, likely lipid pore at DNA lipid interface. Such pores have been demonstrated in experiments and MD simulations, this evidence should not be ignored and the relevant literature should be cited (unless the authors disagree with it, then please discuss in the response).

14) “Current traces of the single LGC-O channels had more high-frequency fluctuations (Fig. 3f, Supplementary Fig. 15b) than the less noisy trace of LGC-C (Fig. 3c, Supplementary Fig. 15a).” The same group published a paper where they attributed the fluctuations to voltage gating at the same applied voltages (Seifert et al., ACS Nano, 2016). This paper is not cited. On purpose? It should not be the case that phenomenologically similar observations are attributed to rational design features if it fits into the story. This is a statement which has to be backed up with a thorough statistical analysis in a revised manuscript. It has to be discussed in detail why the authors once attribute gating to voltage and then to the presence of the flap. Did they not see gating without the flap? I would be surprised if this is the case for such floppy pore walls. They present some evidence by showing that gating depends on whether a positive or a negative voltage is applied. But a real statistical analysis is missing.

Reviewer #2 (Remarks to the Author):

Swarup Dey and coworkers made a DNA sheet that is capable of inserting into lipid bilayers. The DNA structure has a large opening of 20x20 nm at its centre that act as a relatively large nanopore through which molecules can pass through. The aperture can be open and closed by performing a strand displacement reaction.

This is a nice work that shows the potential of DNA nanotechnology. The ability of opening a DNA nanostructure using DNA displacement reaction has been shown before. What is new is the connection of the DNA structure with a vesicle/liposome, which is a favourite delivery vessel for therapy. The ability of controlling the electrical conductance of a nanopore by a DNA strand is perhaps just an academic exercise, but still a good example of controlled transport across nanopores.

The problem I have with this work, is that it is not entirely clear whether the nanopore spans the lipid bilayer and mediates the delivery of molecules. On the one hand, the fluorescence data appear

convincing, although a few questions remain (see later); on the other hand, the electrical recordings are not. In particular, the conductance of the nanopore appears to be far too small to describe a nanopore of 20x20 nm size. A quick look at the literature showed that, typically, 20-nm (circular) nanopore should have a conductance that is at least 10-fold higher. In addition, the conductance of the LGC-C (closed nanopore) is only 2-fold lower than LGC-O (the nanopore is open), and 5-fold lower than the and LGC-N (no lid). Why? Is the DNA very leaky? The authors did not explain. But if it is leaky, why the dyes do not pass through?

Fig. 1dii shows a rather tightly closed central pore. Hence, I would expect a much lower nanopore conductance for LGC-C. Furthermore, assuming that the conductance describes the opening of the nanopore, if the closed nanopore is still one fifth open (which corresponds to an opening of a few nm) the dyes should really pass through in the fluorescence experiment.

Also, the blockade given by trypsin are somehow problematic. They are very long and relatively deep. The dwell time was not given (it should be provided), but it appears hundreds of milliseconds. The protein events show that one third of the current is blocked. Given the current exclusion model, this suggest that the protein should be about 1/3 of the volume of the nanopore, which appears not to be the case, as trypsin is rather small (3-4 nm) compared to the pore (20 nm). I would actually expect very fast and shallow blockades. In fact, very few and fast blockades have been described in the literature for the translocation of proteins across SiNx nanopores of similar size.¹

Therefore, In the present form, I am not convinced that the authors observe the electrophoretic translocation of proteins across the membrane (i.e. I am not convinced that the nanopore connects directly the solutions on both side of the bilayer). A possible explanation for the nanopore recordings is that the authors observe the transport of ions around the dsDNA helices inserted into the bilayer, rather than across the nanopore.² However, if this is the case, it is strange the authors do not observe the leakage of dyes inside the vesicle s. Given that the closed pores are leaky to ions, why the dyes cannot pass?

More experiments should be tested to solve this condrum.

Nanopore experiments:

1. A control experiment using just a DNA sheet including cholesterol and no central aperture. DNA accumulation around the vesicle should be observed in fluorescence experiments, but these structures should provide no conductive pores in lipid biliyers.

2. Perform a voltage dependence of the protein translocation. If proteins translocate through the nanopore, the dwell time should decrease with the applied potential. Provide the dwell times and residual current values (with errors).
3. Test the translocation of GFP
4. Estimate the conductance of the nanopore from its geometry and compare it to the experimental values.

Fluorescence experiments

1. Perform a control experiment using a DNA sheet with cholesterol and no central pore, the authors should see no entry of the dye in fluorescence experiments
2. I would test the exit of the dye from inside the vesicle to the outside solution through LGC-C and LGC-O. This would mimic the release of drugs from vesicles.

Other questions / points

The depiction of the nanopore in the main text is not clear. From Figure 1, the DNA sheet appears to be formed by a single layer of dsDNA. What about the central nanopore? From the figure it appears 3 or four dsDNA strands. However, in Figure S1a the central nanopore appears much longer. Or is this a different nanopore? Why is it there?

Nanopore Design. It is not entirely clear to me how the author established that the central aperture of the DNA plate spans the lipid bilayer. Are the side walls of the central nanopore decorated with cholesterol?

Figure 2b. this figure is not clear. It appears the DNA sheet aggregates on the lipid vesicle

Line 73. Why using 64 cholesterol units? Does the DNA aggregate?

Line 219-220-221-226 etc, please provide the errors.

Line 249. What do the authors mean for dynamically closed nanopore? Please explain.

The experiments with the dynamically open and closed nanopores (Fig. 3 and 4) are not clear. The authors start with a closed pore which does not allow the entry of dyes (dark vesicle). Once the pore is opened the dye can enter (green vesicle). However, when the pore is later closed, why the dyes are not retained inside the vesicle (the vesicle is dark)?

References

1. Plesa, C. et al. Fast translocation of proteins through solid state nanopores. *Nano Lett* 13, 658–663 (2013).
2. Göpfrich, K. et al. Ion Channels Made from a Single Membrane-Spanning DNA Duplex. *Nano Lett.* 16, 4665–4669 (2016).

Reviewer #3 (Remarks to the Author):

In this report, the authors present an attractive large, DNA origami-based, gated nanopore in the artificial lipid membrane capable of protein transportation. DNA nanotechnology has enabled nucleic acid self-assembled transmembrane scaffolds that can functionally mimic protein ion channels. Various DNA origami-based nanopores have been reported previously and characterized. Several works use a ligand such as an oligonucleotide to control the opening and closing of the pores and demonstrate the transportation of biologically important agents across the membrane. The innovation of this work is that the team, for the first time, designed a wide origami pore with a diameter on a scale of 20 nm. This large pore size is in contrast to the reported origami nanopores of several nanometers. Due to the large pore size, the pore demonstrated direct transport of protein such as trypsin. As native biological protein pores cannot form wide channels like this, the proposed origami pore may have new functions superior to the protein channels, with broad applications in biosensing and programmable delivery.

Minor questions

The electrophysiology experiment is the most direct method to prove the origami pore formation (e.g., Fig. 2e, f, and g). One question is that, although Fig. 2e (and several other current traces in supplementary figures) shows a stable origami pore current, the histogram in Fig. 2f shows a broad distribution of the pore conductance. It is not clear whether different origami pores have different conductance levels or the same pore feature various conductance levels. Please illustrate more current traces to clarify this. Furthermore, please explain the conductance diversity of the origami pore. For example, is it possible that the DNA can form different pores as certain staples do not bind as designed?

We thank the reviewers for the helpful feedback. We made changes to the manuscript to implement the reviewers' suggestions and Comments, including stronger evidence that the DNA pore spans the entire membrane and that the molecules are transported through the core in a lid - dependent manner. We provide replies to the reviewers' Comments below:

Reviewers' Comments:

Referee 1 Comments for the Author:

The manuscript by Dey et al. describes a membrane-spanning DNA pore with the largest diameter to date. With a 20nm-wide inner opening, the channel itself is about twice as large as the structurally more stable pore by Thompson et al (Nat. Commun., 2019, ref. 50) with an inner diameter of 9nm. This and other previous pores had the capability to open up in a stimuli-responsive manner by strand displacement. The advance here is that not just opening but also subsequent closure was achieved, although the process is not repeatable. As an additional aspect of novelty, the authors showed transport of a 32kDa protein across the membrane. It should be noted, however, that 40kDa Dextran was transported through DNA nanopores before (Nat. Commun., 2019, ref. 50). In principle, the level of novelty justifies publication – I leave it up to the editor on what level – but the data and the presentation of the results is to large extends questionable. In parts it was difficult to evaluate the manuscript in a fair way because statistical information is missing. Several key aspects have to be addressed thoroughly.

Answer:

We thank the reviewer for appreciating the novelty offered by the current research regarding the reversible gating of a wide nanopore capable of protein transport. The reviewers' Comments about the previous (and some recent) demonstrations of other wide pores as well as some reports showing one-way gating in narrower pores are indeed true. However, we would like to point that the main novelty of the current research is in unifying all the key aspects of a DNA nanopore in a single design toward their application in drug delivery and nanopore sensing. We appreciate the reviewers' suggestions regarding the key experiments and data analysis requirements, crucial to convey the messages claimed in this report. We have addressed and clarified the points raised by the reviewers in the following sections point by point.

Comment:

1) In several instances, one gets the impression that the authors are not quite open about what has already been done to increase the apparent novelty of the current work – partially disregarding their own work. It should be explicitly mentioned that they and others have already demonstrated pore opening by strand displacement and that cargo of similar molecular weight as their protein was already transported through DNA pores.

Answer:

We thank the reviewer for pointing out this and we apologize for giving the impression of disregarding our own work. We have included a detailed discussion in the introduction section regarding how the design and functional aspects of our nanopore are novel compared to the

previous nanopores – both in terms of size as well as the reversible gating. And also mentioned and cited the previous instances of big pore and reversible gating. We think that the major novelty of our nanopore is completely reversible while being wide and stable enough nanopore to translocate larger cargo such as folded proteins.

Comment:

2) The authors state that their pore is 416 nm²-wide. This is misleading, since width is, per definition, not an area. Width is defined as “the measurement or extent of something from side to side; the lesser of two or the least of three dimensions of a body”.

Answer:

The reviewer is indeed correct. We apologize for this mistake. Words like “wide”, “width” has been replaced with “opening cross-section” (lines 25, 74, 156, 161, 174, 572) to correctly represent the term. We chose the cross-sectional area instead of width as width generally gives an impression of a circular cross-section, whereas in this case the pore is a square one.

Comment:

3) The authors currently highlight the reversibility of the pore opening, but do not discuss at all that they currently can only open the pore once and then close it once. This is not a problem, but they should be open about it and maybe suggest strategies to achieve true reversibility.

Answer:

We have repeated the pore opening and closing over 4 cycles (4 openings and 4 closings excluding the starting point) and included the FRET result in the main text Fig. 3 to demonstrate that the pore can indeed be reversibly opened and closed over multiple cycles.

Comment:

4) Minor Comment: From the main it is not so obvious with how many cholesterol-moieties the pore was functionalized. This should be stated in the text and the Fig. caption so that the reader does not have to search in the SI.

Answer:

We previously had the number of cholesterol in the main text (previously line 73, currently line 159). We have mentioned the number of cholesterol in the Fig. 1 caption (currently line 107) as well as per the reviewer’s instructions.

Comment:

5) There is little evidence for the correct formation of the stem part of the pore and there are actually no cholesterol-moieties positioned on the stem. I wonder if it is structurally stable enough to penetrate the membrane when the cholesterol inserts. It is unclear how long the linkers for the chol-DNA are. Removal of a few strands on the stem would allow it to collapse completely. The current data seems to suggest some disturbance of the membrane, but with this concern it has to be interpreted even more carefully.

Answer:

i) We thank the reviewer for raising this important concern if the pore is structurally stable enough to span the membrane. We have performed an orthogonal assay (Supplementary Fig. 11) to cross check this concern. We placed a Cy5-labelled ssDNA probe at the

- bottom edge of the LGC pore and encapsulated Cy3 labelled complementary ssDNA inside GUVs. Cholesterol labelled pores span the bilayer, causing hybridization of the Cy5 labelled probe strand of LGC with the Cy3 labelled strands inside the GUV, establishing that the cholesterol modified LGCs indeed penetrate through the membrane forming a stable transmembrane channel across the bilayer.
- ii) As per the suggestion of reviewer 2, we constructed a flat origami without the stem part (no-cap plate) and performed independent fluorescence (Supplementary Fig. 13) as well as nanopore electrical current recording experiment (Supplementary Fig. 14). These experiments established that the translocation of dye and ions are indeed through the central pore and not through the toroidal pores known to form between the sides of DNA nanopores and the lipid channel.
 - iii) Dye influx experiments have previously been extensively used to demonstrate transmembrane pore formation both in case of DNA nanopores as well as protein and peptide materials. The atto-633 dye and GFP influx experiments together with the supporting experiments above clearly demonstrate the formation of transmembrane pore. This perhaps indicates a more dynamic re-orientation of lipids around DNA nanopores during the formation of the transmembrane channels which are not as dependent on the rigidity of the inserting stem part as thought before in a hole-punching mechanism.
 - iv) The cholesterol-DNA linkers are 22 nucleotides long (Supplementary Table 3.3). The cholesterol is positioned proximal to the plate part of LGC.

Comment:

6) Minor Comment: "To probe for membrane binding, channel variant LGC-N was added to small unilamellar vesicles (SUVs) and analyzed by gel electrophoresis in Mg²⁺ free buffer to avoid non-specific adsorption to chelating lipid head-groups". Have the authors established that the pore is even stable in Mg²⁺-free buffer?

Answer:

We thank the reviewer for pointing this. We have included AFM of the pores in the no-Mg buffer (Supplementary Fig. 8), showing that the pores are indeed stable in the no-Mg buffer and also mentioned this in the main text.

Comment:

7) Minor Comment: Fig. 2c: How large is the scale bar? Put scale bar for confocal in Fig 2c-ii.

Answer:

We have put the scale bars (10 μm) now.

Comment:

8) Fig. 2d shows a bar plot of the percentage of GUVs showing a filled interior after 3 hours. It is unclear how the data was analyzed and how the authors distinguished a vesicle with a filled interior from an unfilled interior. It would be helpful to include a plot with more information,

namely to plot the Intensity inside the vesicle over the Intensity of the surrounding. What do the authors mean by percentage of influx?

Answer:

We apologize for not making it clear earlier. We have replotted the bar plot, implementing the changes suggested by the reviewer. The mean intensity inside each GUV in a frame was measured using ImageJ software. For each experiment, the intensity inside the GUVs at $t = 0$ min (I_0) and at time, t (I_t) was measured and normalized by average intensity of the background at the same time point as $I_{t,norm} = \frac{I_{t,i}}{I_{t,b}}$; where $I_{t,norm}$ = normalized GUV interior intensity at time t ; $I_{t,i}$ = GUV interior intensity at time t and $I_{t,b}$ = average background intensity at time t . GUVs that satisfied the threshold criteria $(I_{t,norm} - I_{0,norm}) > 0.5$ was considered to have shown influx at that time point and from this the percentage of GUVs showing influx was calculated. This has been stated in the Materials and Methods section in manuscript.

Comment:

9) The authors explicitly highlight the faster influx kinetics compared to the previous publication of Thompson et al. but they never show the influx over time (even though this should be straight forward since they monitor the process with fluorescence microscopy). They explain the faster kinetics as follows: “The faster flux is likely due to the larger pore lumen of LGC, the higher number of our inserted channels, or a combination of both.” Have they checked the plausibility of this statement? My guess is that for a pore of that size, one would expect much faster equilibration once it is inserted. The authors should do some back of the envelope calculations and show the individual influx traces.

Answer:

We thank the reviewer for raising this point. We have added the single traces obtained from the confocal micrographs (Supplementary Fig. 12) and added a Supplementary note 1 for a rough calculation of the theoretical influx times. We have also elaborated our discussion in the main text explaining our 3 unique observations – a) We observed much higher % of the GUVs getting influx compared to previous reports by Thompson et al¹ and Iwabuchi et al². b) The dye influx completion time in our case (~1h) was much faster compared to that observed by other reports and c) The observed dye influx rate is ~5X slower than the theoretically predicted value considering the pore dimensions.

The influx % observed with our LGC pores is significantly higher compared to that of 11% in case of Thompson et al and 8-10% observed by Iwabuchi et al. Moreover, the LGC mediated influx completed within 1h after channel addition, which is considerably faster than 5-8h completion time required in case of previous DNA nanopores. The observed rate of dye diffusion through DNA nanopore is a combination of several inter-dependent phenomena, i.e.- i) nanopore binding to the membrane, ii) reorientation of its stem in the bilayer to form a channel and finally iii) the diffusion of the dye through the pore. The nanopore binding to membrane is strongly governed by the number of cholesterol molecules, ionic concentration, temperature etc. While

the reorientation step is supposedly the slowest and rate determining step that depends on the flexibility of the cholesterol bearing segments of the nanopore. The final step of dye diffusion is a function of the size of dye molecule, medium viscosity, temperature, concentration gradient and pore dimensions. We attribute the faster flux in case of our LGC pore compared to the previous pores due to the placement of large number of cholesterol anchors to help the step (i). The cholesterol-bearing one layer origami plate being flexible enough for rapid re-orientation of the pores, probably led to faster insertion kinetics compared to the previous origami pores that always placed the cholesterol molecules in a rigid framework, thereby slowing the reorientation step. Interestingly, the dye diffusion rate in LGC pore was ~5 times slower compared to the theoretically calculated time required for diffusion (Supplementary Fig. 12 & Supplementary note 1). This might have resulted as contribution of the wider pore dimension of LGC toward the diffusion rate was overcome by the reorientation step that governs the overall observed rate of dye influx. Further detailed investigation is required to decouple the absolute contributions of the three steps in the insertion of hydrophobically modified DNA nanostructures into lipid bilayer.

Comment:

10) I am very surprised how messy the GUV preparation looks. I could well imagine that this is due to the inverted emulsion method. Is it possible that the authors did not control the humidity in the lab? Since encapsulation is not needed in this case, it would be much preferred to use the electroformation method. This avoids oil traces and should give much cleaner results.

Answer:

We thank the reviewer for suggesting this. The messy look of the GUVs come from the debris settled across the 3 hours course of imaging. In our experience we have found it very challenging to remove these debris by either electroformation as well as inverted emulsion method. However, this did not affect the unilamellarity of the GUVs. Most of the GUVs that looked messy were brightfield images in the SI, showing proper focus during imaging. Since the fluorescence channels are enough to show the proper focus we have removed the brightfield images from the SI.

Comment:

11) I am somewhat surprised that the authors claim that Atto633 cannot diffuse through the closed pore. A single layer of DNA is porous itself and I would expect that a small dye molecule may well be able to penetrate through. Additionally, it is known that DNA pores form a toroidal pore at the DNA-lipid interface which should transport small dye molecules – especially for this pore which has no hydrophobic entities on the stem. Can the authors discuss why this should not be the case? It would be ideal to proof it with an independent experiment. Have the authors looked at longer time scales? Especially since the residual current of the closed pore is still half of the current of the open pore, I would not expect complete blockade of the fluorophore transport.

Answer:

We have repeatedly observed that the small molecule Atto633 dye does not diffuse through the closed lid pore. We have done independent experiment as per the reviewer 2's suggestion where we constructed a flat origami without the stem part and performed independent fluorescence (Supplementary Fig. 14) as well as nanopore electrical current recording experiment (Supplementary Fig. 15). These ruled the possibility of translocation of dye and ions through the toroidal pores.

Moreover, Atto633 dye has a molecular dimension of $\sim 1.3\text{nm}$ which is greater than the Debye length of DNA helices ($\approx 0.5\text{nm}$) in DNA origami (Wook & Rothmund, Nat Comm 2014). Thus, it is not surprising that the Atto633 dye does not pass through the single layered lid.

Next, the lack of dye translocation and the leakage of ions through the lid in the closed state cannot be compared due to the following reasons: i) The ions being much smaller ($0.1\text{-}0.2\text{nm}$) compared to the Atto633 dye molecule (1.3nm) has much higher permeability. ii) The conditions of the fluorescence and the nanopore current recording experiments are not identical. The nanopore current recording experiments are done under an applied electric field that can partially distort the single layered lid. Whereas such distortions are unlikely to happen in fluorescence microscopy experiments that are performed in neutral, native conditions. Thus, we believe while the LGC channels are not completely ion-tight, they are very tight with respect to small molecule cargo release.

Comment:

12) *“These data clearly confirm that the lid can be dynamically opened and closed specially with external triggers, something which has not be achieved before by any means.” Since opening has been achieved, this is a misleading statement. Restate with emphasis of closing back and reversibility of the lid.*

Answer:

We thank the reviewers for pointing this out. We have shown reversible opening and closing over several cycles in the revised manuscript. And the above mentioned sentence has been rephrased accordingly with special emphasis on reversibility.

Comment:

13) *“The small residual current of LGC-C might stem from ion leakage either through the periphery of the lid or through the DNA duplexes of the lid, or both.” The small residual current is not small at all, it is half of the open current, likely lipid pore at DNA lipid interface. Such pores have been demonstrated in experiments and MD simulations, this evidence should not be ignored and the relevant literature should be cited (unless the authors disagree with it, then please discuss in the response).*

Answer:

We thank the reviewer for pointing this out. We indeed believe that part of the residual current from the LGC-C stems from a toroidal pore formation at the duplex/lipid interface. We have updated the text to reflect this and cited some key literature that shows this very effect. Whilst we believe this is a factor, there is still the possibility of ion permeability through the lid – most likely a combination of these two effects results in the residual currents we see with LGC-C. Supplementary Fig. 6 (below) shows oxDNA simulations of LGC-N, LGC-C and LGC-O, this Fig. can maybe help illustrate the reason for a) currents in LGC-C (through the periphery of the lid) and b) the reason for a relatively small open current in LGC-O in comparison to LGC-N, as the lid is still partially blocking the pore lumen.

Comment:

14) “Current traces of the single LGC-O channels had more high-frequency fluctuations (Fig. 3f, Supplementary Fig. 15b) than the less noisy trace of LGC-C (Fig. 3c, Supplementary Fig. 15a).” The same group published a paper where they attributed the fluctuations to voltage gating at the same applied voltages (Seifert et al., ACS Nano, 2016). This paper is not cited. On purpose? It should not be the case that phenomenologically similar observations are attributed to rational design features if it fits into the story. This is a statement which has to be backed up with a thorough statistical analysis in a revised manuscript. It has to be discussed in detail why the authors once attribute gating to voltage and then to the presence of the flap. Did they not see gating without the flap? I would be surprised if this is the case for such floppy pore walls. They present some evidence by showing that gating depends on whether a positive or a negative voltage is applied. But a real statistical analysis is missing.

Answer:

We thank the reviewer for spending the time to compare our data to previous publications in the field. The gating seen in (Seifert et al., ACS Nano, 2016) are vastly different to the gating effects we see with LGC-O. The ACS Nano pore is a small 6-helical bundle structure with no lid that becomes dramatically blocked (almost completely) at high voltages (>80 mV) (see below). These blocks are long lasting (tens of seconds) and do not occur at low voltages. Indeed, our gating effects are more closely related to that of (Burns et al., Nat Nano, 2016). In this paper a similar 6-helical bundle structure is presented, this time with strands present that hold the lid in place. When the lid is not present these strands are able to partially block the pore. The blocking events are rapid (μs – ms range) and occur at both low and high voltage. The gating events seen in LGC-O are also rapid (μs – ms range) and occur at both low and high voltages. We have also added some additional analysis comparing the noise of LGC-O and LGC-C at all voltages (Supplementary Fig. 22) and also comparing LGC-O noise and blocking events at positive and negative membrane potentials (Supplementary Fig. 23 and 24).

[Redacted]

[Redacted]

Reviewer #2 (Remarks to the Author):

Swarup Dey and coworkers made a DNA sheet that is capable of inserting into lipid bilayers. The DNA structure has a large opening of 20x20 nm at its centre that act as a relatively large nanopore through which molecules can pass through. The aperture can be open and closed by performing a strand displacement reaction.

This is a nice work that shows the potential of DNA nanotechnology. The ability of opening a DNA nanostructure using DNA displacement reaction has been shown before. What is new is the connection of the DNA structure with a vesicle/liposome, which is a favourite delivery vessel for therapy. The ability of controlling the electrical conductance of a nanopore by a DNA strand is perhaps just an academic exercise, but still a good example of controlled transport across nanopores.

The problem I have with this work, is that it is not entirely clear whether the nanopore spans the lipid bilayer and mediates the delivery of molecules. On the one hand, the fluorescence data appear convincing, although a few questions remain (see later); on the other hand, the electrical recordings are not.

Comment:

In particular, the conductance of the nanopore appears to be far too small to describe a nanopore of 20x20 nm size. A quick look at the literature showed that, typically, 20-nm (circular) nanopore should have a conductance that is at least 10-fold higher. In addition, the conductance of the LGC-C (closed nanopore) is only 2-fold lower than LGC-O (the nanopore is open), and 5-fold lower than the and LGC-N (no lid). Why? Is the DNA very leaky? The authors did not explain. But if it is leaky, why the dyes do not pass through? Fig. 1dii shows a rather tightly closed central pore. Hence, I would expect a much lower nanopore conductance for LGC-C. Furthermore, assuming that the conductance describes the opening of the nanopore, if the closed nanopore is still one fifth open (which corresponds to an opening of a few nm) the dyes should really pass through in the fluorescence experiment. Difference between electrical conductance and dye influx has to be explained. In terms of difference between ion and dye sizes and/or behavior of negatively charged, floppy lid in presence (for electrical recordings) and absence (for fluorescence experiments) of electric field.

Answer:

We thank the reviewer for taking the time to thoroughly look through our electrophysiological data. Indeed, the pore conductance of LGC-N is smaller than predicted, this has been seen before with large DNA nanopores in the literature (Diederichs *et al.*, Nat Comm, 2019) and through my experience working with large diameter DNA nanopores (currently unpublished data). This could be due to compression of the nanopores when inserted into lipid bilayers or due to the high density of negative charge caused by the DNA phosphate backbone – which could act to hinder

ion flow. We believe that part of the residual current from the LGC-C stems from a toroidal pore formation at the duplex/lipid interface. We have updated the text to reflect this and cited some key literature that shows this effect. We also believe that ion permeability through the lid also plays a role. Toroidal pore formation at the duplex/lipid interface is large enough for single ions to flow through, but not for dye flow as the dye molecules are significantly larger than single ions. For example, a chloride ion has an ionic radius of 0.18 nm, whilst the diameter of a typical dye chromophore is 1 – 1.5 nm, a near tenfold difference in size. For this reason, we would not expect that dye can permeate just because ions can, and indeed this is demonstrated by the data. Supplementary Fig. 6 (below) shows oxDNA simulations of LGC-N, LGC-C and LGC-O, this Fig. can maybe help illustrate the reason for a) currents in LGC-C (through the periphery of the lid) and b) the reason for a relatively small open current in LGC-O in comparison to LGC-N, as the lid is still partially blocking the pore lumen.

Comment:

Also, the blockade given by trypsin are somehow problematic. They are very long and relatively deep. The dwell time was not given (it should be provided), but it appears hundreds of

milliseconds. The protein events show that one third of the current is blocked. Given the current exclusion model, this suggests that the protein should be about 1/3 of the volume of the nanopore, which appears not to be the case, as trypsin is rather small (3-4 nm) compared to the pore (20 nm). I would actually expect very fast and shallow blockades. In fact, very few and fast blockades have been described in the literature for the translocation of proteins across SiNx nanopores of similar size.1

Answer:

We thank the reviewer for their Comments. We have now repeated the trypsin translocation experiments and provided a more complete analysis with improved presentation (Fig. 4b). This new analysis is in line with previous trypsin block events through a DNA nanopore (Diederichs *et al.*, Nat Comm, 2019)⁵. The average block of ~18% suggests a block of around 1/5 of the nanopore volume, roughly in line with the 20 nm width of the DNA nanopore and 3-4 nm diameter of trypsin. As is observed with the smaller than expected LGC-N currents, the pore is likely compressed in the lipid bilayer, which would result in trypsin blocking to a larger extent than expected.

Comment:

Therefore, In the present form, I am not convinced that the authors observe the electrophoretic translocation of proteins across the membrane (i.e. I am not convinced that the nanopore connects directly the solutions on both side of the bilayer). A possible explanation for the nanopore recordings is that the authors observe the transport of ions around the dsDNA helices inserted into the bilayer, rather than across the nanopore.2 However, if this is the case, it is strange the authors do not observe the leakage of dyes inside the vesicle s. Given that the closed pores are leaky to ions, why the dyes cannot pass? More experiments should be tested to solve this conundrum.

Answer:

Many thanks for your Comment. If it was the case that the nanopores do not connect the solutions on both sides of the bilayer, then all three constructs (LGC-C, LGC-O and LGC-N) would be expected to give identical conductance characteristics. As this is not the case the more plausible explanation is that indeed the nanopores do puncture the membrane and allow for current flow through the lumen – resulting in vastly different conductance characteristics of the three pore types. In addition to this, repeated trypsin translocation data and new GFP translocation data giving rise to blocking events further proves that indeed these pores are permeating the membrane – as translocation only occurs when using LGC-N, and not with LGC-C (Supplementary Fig. 31). As mentioned above, a chloride ion has an ionic radius of 0.18 nm, whilst the diameter of a typical dye chromophore is 1 – 1.5 nm, a near tenfold difference in size.

For this reason, we would not expect that dye can permeate just because ions can, and indeed this is highlighted in the data.

Comment:

Nanopore experiments:

1. A control experiment using just a DNA sheet including cholesterol and no central aperture. DNA accumulation around the vesicle should be observed in fluorescence experiments, but these structures should provide no conductive pores in lipid bilayers.

Answer:

We thank the reviewer for the suggestion. These experiments have been conducted, and as expected the DNA plates do not form conductive pores in lipid bilayers (Supplementary Fig. 14).

Comment:

2. Perform a voltage dependence of the protein translocation. If proteins translocate through the nanopore, the dwell time should decrease with the applied potential. Provide the dwell times and residual current values (with errors).

Answer:

We thank the reviewer for the suggestion. These experiments have been carried out (Supplementary Fig. 32) and indeed dwell time does decrease with an increase in the applied potential. Fig 4b has also been updated to show a more complete analysis of trypsin translocation with regards to channel block and dwell times. The main text has been updated to show average blocks and dwell times with errors.

Comment:

3. Test the translocation of GFP

Answer:

We thank the reviewer for their suggestion. Translocation studies have now been carried out using GFP (Supplementary Fig. 33 and 34). GFP translocation data is comparable to previous

studies using DNA nanopores (Diederichs *et al.*, Nat Comm, 2019)⁵ – occurring only once GFP is in a net positive conformation (at low pH). GFP translocation gives a larger extent of blocking than trypsin, in line with the differences in hydrodynamic radii (~3 nm for trypsin and 5.6 nm for GFP) highlighting that translocation events are due to protein translocation.

Comment:

4. Estimate the conductance of the nanopore from its geometry and compare it to the experimental values.

Answer:

Using the equation:

$$G = \kappa \frac{\pi d^2}{4L + \pi d}$$

Gives a predicted conductance of 132 nS. This equation doesn't factor in compression of the nanopore, or the charge of the DNA duplexes, and as such is not an accurate estimate for predicting conductance of DNA nanopores. Comparing LGC-N with a previous large diameter DNA nanopore (Diederichs *et al.*, Nat Comm, 2019)⁵ we can see that there is an increase of roughly 3x – in line with an increase in pore diameter from 7 of the previously reported pore to 20 nm of the pore in this manuscript, suggesting that the experimental conductances are consistent between DNA nanopores.

Fluorescence experiments

Comment:

1. Perform a control experiment using a DNA sheet with cholesterol and no central pore, the authors should see no entry of the dye in fluorescence experiments.

Answer:

We thank the reviewer for suggesting this experiment. We have performed this experiment and included this in the SI (Supplementary Fig. 13).

Comment:

2. I would test the exit of the dye from inside the vesicle to the outside solution through LGC-C and LGC-O. This would mimic the release of drugs from vesicles.

Answer:

We thank the reviewer for suggesting this experiment. We have performed this experiment and included this in the SI (Supplementary Fig. 29).

Other questions / points

Comment:

The depiction of the nanopore in the main text is not clear. From Fig. 1, the DNA sheet appears to be formed by a single layer of dsDNA. What about the central nanopore? From the Fig. it appears 3 or four dsDNA strands. However, in Fig. S1a the central nanopore appears much longer. Or is this a different nanopore? Why is it there?

Answer:

We are sorry to know that the depiction of the nanopore was not clear. The reviewer is correct that the sheet (or plate) is indeed formed by a single layer of dsDNA. Similarly, the central nanopore is also formed by 3 dsDNA strands excluding the dsDNA of the plate, stacked on top of each other. The longer nanopore in Fig. S1a depicts previous nanopore from Diederichs et al, Nat Comm, 2019.⁵ This pore is put there to show the critical design novelties devised in this report that helped us achieve the superior functionality – more available surface for cholesterol placement while accommodating a lid. These design aspects were previously unachievable which limited the creation of a big nanopore with a lid and enough surface for lot of cholesterol placement. These are described in the caption as well as the “Design of a large and gated DNA channel” subsection in the “Results” section of the main text.

Comment:

Nanopore Design. It is not entirely clear to me how the author established the central aperture of the DNA plate spans the lipid bilayer. Are the side walls of the central nanopore decorated with cholesterol?

Answer:

The reviewer indeed raises a valid point here. This point was also raised by reviewer 1.

- i) Dye influx experiments have previously been extensively used to demonstrate transmembrane pore formation both in case of DNA nanopores as well as protein and peptide materials.^{6,7} The Atto633 dye and GFP influx experiments together with the supporting experiments above clearly demonstrate the formation of transmembrane pore. This perhaps indicates a more dynamic re-orientation of lipids around DNA nanopores during the formation of the transmembrane channels which are not as dependent on the rigidity of the inserting stem part as thought before in a hole-punching mechanism.
- ii) We have performed an orthogonal assay (Supplementary Fig. 11) to cross check this concern. We placed a Cy5-labelled ssDNA probe at the bottom edge of the LGC pore and encapsulated Cy3 labelled complementary ssDNA inside GUVs. Cholesterol labelled pores span the bilayer, causing hybridization of the Cy5 labelled probe strand of LGC with the Cy3 labelled strands inside the GUV, establishing that the cholesterol modified

LGCs indeed penetrate through the membrane forming a stable transmembrane channel across the bilayer.

- iii) Independent fluorescence experiment (Supplementary Fig. 13) as well as nanopore electrical current recording experiments (Supplementary Fig. 14) with the flat origami without the stem part established that the translocation of dye and ions are indeed through the central pore and not through the toroidal pores known to form between the sides of DNA nanopores and the lipid channel.

These points together establish the formation of a membrane spanning pore by LGC.

Comment:

Fig. 2b. this Fig. is not clear. It appears the DNA sheet aggregates on the lipid vesicle.

Answer:

The TEM images in the Fig. 2b show interaction of cholesterol modified LGC pores with the small unilamellar vesicle liposomes. Blue arrowheads pinpoint the pores where the central square pore is visible. The spherical structures represent the liposomes.

Comment:

Line 73. Why using 64 cholesterol units? Does the DNA aggregates?

Answer:

We thank the reviewer for asking this question. According to Thompson *et al*, Nat Comm, 2019's continuum model¹ and MD simulation predictions (Supplementary Fig. 2, Thomson et al, Nat Comm 2019) a pore of ~20nm width would require > 60 cholesterol probes. Thus, we used 64 cholesterol probes. We have also included this in Supplementary Fig. 2 now.

Comment:

Line 249. What do the authors mean for dynamically closed nanopore? Please explain. The experiments with the dynamically open and closed nanopores (Fig. 3 and 4) are not clear. The authors start with a closed pore which does not allow the entry of dyes (dark vesicle). Once the pore is opened the dye can enter (green vesicle). However, when the pore is later closed, why the dyes are not retained inside the vesicle (the vesicle is dark)?

Answer:

We apologize for the confusion caused here. We show the reversible opening of the closed nanopore (we call this dynamic opening) by adding the opening key to the solution of GUV,

LGC-C and Atto633 dye (Fig. 3b – middle panel). However, we want to show similar dynamic closing in this set up by adding closing key to this solution, it causes too much disturbance in osmolarity which causes the vesicles to rupture. To bypass this problem, we separately form a “dynamically closed pore” which we obtain by first, dynamically opening the closed LGC by addition of opening key, followed by addition of closing key. And then we add this “dynamically closed pore” to fresh GUVs that does not contain any dye inside like the set up with closed pore as shown in Fig. 3b – top panel. That is why the interior of the vesicle look black. We understand that this is confusing, thus we added additional cartoons to the Fig. showing a dropper that denotes at which step the dye was added. And add these details to the caption.

Our FRET experiments indeed demonstrate that the LGC lid can be reversibly opened and closed over several cycles. The inability to show the in the confocal experiment is purely due to experimental challenges. Moreover, we performed a two-color confocal microscopy experiment (Supplementary Fig. 20) where first a red dye (Atto 633) is added to the LGC-O. Due to the open pore the dye is able to influx into the vesicle. Later the closing key is added only to the bottom panel. In the top panel the pore remains open while in the bottom panel the closing key renders the pore closed. A green dye (Atto 488) was then added to both samples. The dye only influxes into the vesicle where the pore is open.

Reviewer #3 (Remarks to the Author):

In this report, the authors present an attractive large, DNA origami-based, gated nanopore in the artificial lipid membrane capable of protein transportation. DNA nanotechnology has enabled nucleic acid self-assembled transmembrane scaffolds that can functionally mimic protein ion channels. Various DNA origami-based nanopores have been reported previously and characterized. Several works use a ligand such as an oligonucleotide to control the opening and closing of the pores and demonstrate the transportation of biologically important agents across the membrane. The innovation of this work is that the team, for the first time, designed a wide origami pore with a diameter on a scale of 20 nm. This large pore size is in contrast to the reported origami nanopores of several nanometers. Due to the large pore size, the pore demonstrated direct transport of protein such as trypsin. As native biological protein pores cannot form wide channels like this, the proposed origami pore may have new functions superior to the protein channels, with broad applications in biosensing and programmable delivery.

Comment:

Minor questions

The electrophysiology experiment is the most direct method to prove the origami pore formation (e.g., Fig. 2e, f, and g). One question is that, although Fig. 2e (and several other current traces in supplementary Fig.s) shows a stable origami pore current, the histogram in Fig. 2f shows a

broad distribution of the pore conductance. It is not clear whether different origami pores have different conductance levels or the same pore feature various conductance levels. Please illustrate more current traces to clarify this. Furthermore, please explain the conductance diversity of the origami pore. For example, is it possible that the DNA can form different pores as certain staples do not bind as designed?

Answer:

We would like to thank the reviewer for taking the time to read through the paper and providing their feedback. With regards to the conductance range of LGC-N, this is due to different conductance states of the pore – not due to the same pore having various conductance levels. This has been previously reported for large diameter DNA nanopores (Diederichs *et al.*, *Nat Comm*, 2019)⁵. For clarity we have added an additional Fig. to the SI (Supplementary Fig. 15) that shows an example of six individual pore insertions of LGC-N. This Fig. shows how pores can either be low conductance (<5 nS) or high conductance (>5 nS) – likely due to small differences in structural shape of the lumen (as can be seen in Fig 1d-i).

REFERENCES

- 1 Thomsen, R. P. *et al.* A large size-selective DNA nanopore with sensing applications. *Nat. Commun.* **10**, 5655, (2019).
- 2 Iwabuchi, S., Kawamata, I., Murata, S. & Nomura, S.-i. M. A large, square-shaped, DNA origami nanopore with sealing function on a giant vesicle membrane. *Chemical Communications* **57**, 2990-2993, (2021).
- 3 Seifert, A. *et al.* Bilayer-spanning DNA nanopores with voltage-switching between open and closed state. *ACS Nano* **9**, 1117-1126, (2015).
- 4 Burns, J. R., Seifert, A., Fertig, N. & Howorka, S. A biomimetic DNA-based channel for the ligand-controlled transport of charged molecular cargo across a biological membrane. *Nature Nanotechnology* **11**, 152-156, (2016).
- 5 Diederichs, T. *et al.* Synthetic protein-conductive membrane nanopores built with DNA. *Nat. Commun.* **10**, 5018, (2019).
- 6 Plesa, C. *et al.* Fast translocation of proteins through solid state nanopores. *Nano Letters* **13**, 658-663, (2013).
- 7 Göpfrich, K. *et al.* Ion channels made from a single membrane-spanning DNA duplex. *Nano Letters* **16**, 4665-4669, (2016).

REVIEWER COMMENTS

Reviewer #1 (Remarks to the Author):

The authors have addressed my comments adequately.

Reviewer #2 (Remarks to the Author):

The authors addressed my points and performed new experiments. At this point, I am still not entirely convinced that the DNA nanopores allow the analysis of large proteins. This is because the conductance of the nanopores are much smaller (by 25-fold!) than what is expected for the size of the nanopore designed. The authors refer to previous literature with other DNA nanopores. However, this is a circular argument, as it could simply indicate that DNA is not forming conventional pores. Furthermore, to argue for the discrepancy in size / conductance, the authors indicate that DNA might reduce current flow because it is charged. But why would that be? If anything, a charged surface should increase the current flow, not decrease it.

The observation that protein induces large blockades is also suggestive that the nanopores do not assemble as expected by the authors, and are much smaller. Free protein diffusion is too fast to be observed by nanopore recordings. In fact, it has been widely reported that protein blockades only appear only if there is an interaction between the nanopore and the protein, or if the nanopore inner diameter is similar to the size of the protein. Furthermore, the concentration of trypsin used is very large (38 μM). This is 1000-fold higher than the typical concentration used in experiments with biological nanopores (which have much smaller diameter and thus capture radius). Why is this the case?

In my opinion, it is unlikely the ionic current can pass through the nanopore from its central aperture. However, this should not prevent publication. I only recommend the authors to make this point clear in the text so a reader can make up her mind whether these pores assemble in lipid bilayers as expected, if at all.

More detailed comments.

Line 185. Here the authors should comment on the discrepancy between the current measured and expected size of the nanopore. In the rebuttal, the author calculates a predicted conductance of 132 nSi, which is 26-fold lower than observed experimentally (5 nSi)! Here, in the main text the authors should address this discrepancy and clearly state what is the size of the nanopore as calculated from the conductance. Then, the authors should make their arguments why they think there is this discrepancy. In order to avoid many back-and-forth arguments, I recommend the authors to also provide a convincing argument (backed up with references and/or MD simulations) why DNA would provide such a reduced ionic flow.

Line 257. Please explain why you expect that the closed nanopore still carries a rather large ionic current

Line 283. Please indicate the applied potential.

Line 333. Please indicate the sampling and filtering frequencies. From the data shown in Fig. 4, bii and biv, it appears the events are largely under sampled. Please notice that in order to have meaningful rates, the events should be oversampled (e.g. the sampling rate should be at least 50 kHz and filtered at 10 kHz). [line 493, what is the meaning for having a not Bessel filter?] Although it appears from methods sampling might be correct, the data suggest that the events were sampled at lower frequency. From Fig. b(iv) the sampling rate appears 10 kHz. Fig 4bii suggest a sampling rate of 10 kHz. With a 50 kHz sampling frequency the event is therefore $\sim 50 \mu\text{s}$, rather than the $\sim 250 \mu\text{s}$. Please show (and collect) data at 50 kHz sampling rate.

Line 353. Please indicate on which side the protein was added.

Line 355. I think the author refers to SI Fig 32 rather than 31. Other numbering appear incorrect. Please check the numbering of all the SI figures. Please indicate the concentration of trypsin used.

Line 368. Please define what is 3.8% and 18%.

Line 371. Please indicate the concentration of GFP.

We thank you and the reviewers for the helpful feedback. We made changes to the manuscript to implement the reviewers' suggestions and Comments. We provide replies to the reviewers' Comments below:

Reviewer rebuttal

Referee 1 Comments for the Author:

Comment:

The authors have addressed my comments adequately.

Answer:

We thank the reviewer for taking the time to read through the revisions and appreciate the comment.

Referee 2 Comments for the Author:

Comment:

1) The authors addressed my points and performed new experiments. At this point, I am still not entirely convinced that the DNA nanopores allow the analysis of large proteins. This because the conductance of the nanopores are much smaller (by 25-fold!) than what is expected for the size of the nanopore designed. The authors refer to previous literature with other DNA nanopores. However, this is a circular argument, as it could simply indicate that DNA is not forming conventional pores. Furthermore, to argument for the discrepancy in size / conductance, the authors indicate that DNA might reduce current flow because it is charge. But why would that be? If anything, a charged surface should increase the current flow, not decrease it.

Answer:

Many thanks for taking the time to read through our revised manuscript. Indeed, it is likely that the experimental pore size is smaller than expected, and we have now added a new section at the end of the results section discussing the advantages and limitations of the pore design. We stress, however, that all experimental data support the controlled transport of protein across the DNA channel lumen. The experimental evidence is also summarized in the new part at the end of the results section.

With regards to reduced current flow of a charged DNA surface, this is possible due to electroosmotic flow. DNA has a net negative charge, which can develop a double layer of cations attracted to it. The inner cation layer will be stationary, whilst the outer layer is free to move. The potential difference (zeta potential) present between this double layer is a cause of electroosmotic flow, as the positive charge of the liquid column can cause the entire solution to be pumped towards the cathode. As a direct result of this, analysis times for cations are shortened, whilst anions can be drastically slowed or even reversed under these conditions¹. In terms of effective current output, this electroosmotic flow could cause smaller than expected currents when measuring DNA nanopores if a subset of ions isn't flowing in the direction of electrophoretics (see figure below from <https://www.kapillarelektrophorese.com/theory/electroosmotic-flow/>). It is unlikely that electroosmotic flow can account for the entire extent of conductance reduction, but this could be a factor coupled with a compressed or distorted pore lumen^{2,3}, or angled insertions of the pore in the membrane⁴.

[Redacted]

Another possibility to consider is the access resistance of the system. Access resistance is the resistance of the medium outside both pore entrances. It has been previously noted that the effect of access resistance is significant for low thickness-to-diameter ratios, i.e. nanopores which lengths are smaller than their diameter⁵ – as is the case for our nanopore. More specifically, the access resistance becomes the dominant component of pore resistance (over the resistance of the pore itself) when $L/a < 1.57$ (L = length of pore, a = pore radius)⁶. In the case of our nanopore $L/a = \sim 0.8$, meaning that the access resistance will be the dominant form of resistance in our system. This is without taking into consideration the charge of the pore itself, which has been shown to increase the ratio of access resistance to pore resistance – suggesting that the role of access resistance becomes more important⁵. A higher access resistance means a reduction in ion flow through the conductor, which again could be a factor in the reduced conductance of our nanopores.

Comment:

2) The observation that protein induce large blockades is also suggestive that the nanopores do not assemble as expected by the authors, and are much smaller. Free protein diffusion is too fast to be observed by nanopore recordings. In fact, it has been widely reported that protein blocked only appear if there is an interaction between the nanopore and the protein, or if the nanopore inner diameter is similar to the size of the protein. Furthermore, the concentration of trypsin used is very large (38 μM). This is 1000-fold higher than the typical concentration used in experiments with biological nanopore (which have much smaller diameter and thus capture radius). Why is this the case?

Answer:

Many thanks for your comments and questions. Indeed, the pore blocks by trypsin and GFP in correlation with the reduced conductance do suggest a smaller pore size. We have added a section at the end of the results section discussing the advantages and limitation of the pore design. Based on calculations of trypsin and GFP volume (34 nm^3 and 92 nm^3 respectively, corresponding to hydrodynamic radii of 2.0 and 2.8) current blockades of 11% and 26% suggest pore volumes of 309 nm^3 and 353 nm^3 respectively. These values correlate to pore diameters of 6.2 nm and 6.6 nm which is smaller than the initial pore design. But the diameters are large enough to accommodate transport of folded protein which was the main aim of this paper. These values may underestimate the actual pore diameter when taken into consideration with the points on reduced current flow above.

The reviewer is indeed correct that free protein diffusion is too fast to be observed by nanopore recordings, which is why we suggest that only positively charged protein conformations will be detected using DNA nanopores – due to electrostatic interactions between the charged protein and the DNA backbone. This slowed transport via pore interaction is shown in our study using GFP in pH buffers above the pI of GFP. Here the GFP is in a net negative confirmation, and under these conditions we do not observe translocation events. But we can observe GFP translocation in pH buffers under the pI of GFP and the protein is positively charged. This is a result we have seen previously⁷. The increased dwell time of positively charged protein through a DNA nanopore has also been reported in modelling studies⁸.

In relation to this, only proteins that interact with the nanopore upon translocation will be sufficiently slowed and detected by the electrophysiological equipment. Hence, a larger concentration is required for detection. We have now added a concentration dependence of trypsin translocation (Supp Fig 33), which shows how the events detected per second are low at trypsin concentrations of <100 nM, and higher at μM concentrations. As there are more events to analyse, we have chosen μM concentrations for characterization of translocation.

Comment:

3) In my opinion, it is unlikely the ionic current can pass through the nanopore from its central aperture. However, this should not prevent from publication. I only recommend the authors to

make this point clear in the text so a reader can make up her mind whether these pores assemble in lipid bilayers as expected, if at all.

Answer:

We thank you for your comments. We have now added a section on the advantages and limitations of the pore design to the end of the results section as a way for readers to make up their own mind, whilst also allowing future researchers to take some of these limitations into consideration when designing any next generation DNA nanopores/nanostructures. We do however refute the idea that it is unlikely that ionic current can pass through the central aperture, as this fundamentally disregards all evidence in this paper (electrophysiological and fluorophore flux). As GFP and dyes are able to pass only through open lid nanopores in the vesicle transport assays, then ions many orders of magnitude smaller will also be able to pass. Closed lid pores in SUV studies do not pass dye or GFP and so an open central aperture is essential for both dye and GFP flux. In further support of transport through the central channel, closed lid nanopores in the single-channel current recordings give a background current almost an order of magnitude smaller (0.64 nS) compared to pores that simply have the central aperture open (5.94 nS) with no other changes to the pore structure. If ions could not flow through the central aperture -as claimed by the reviewer- then both pores would likely give similar conductances.

Comment:

4) Line 185. Here the authors should comment on the discrepancy between the current measured and expected size of the nanopore. In the rebuttal, the author calculates a predicted conductance of 132 nSi, whis is 26-fold lower than observed experimentally (5 nSi)! Here, in the main text the authors should address this discrepancy and clearly state what is the size of the nanopore as calculated from the conductance. Then, the authors should make their arguments why they think there is this discrepancy. In order to avoid many back-and-forth arguments, I recommend the authors to also provide a convincing argument (backed up with references and/or MD simulations) why DNA would provide such a reduced ionic flow.

Answer:

Many thanks for your comments. We have now added a section at the end of the results section that explains the advantages and limitations of the pore design, highlighting the discrepancies in conductance – giving suitable suggestions in relation to the literature.

Comment:

5) Line 257. Please explain why you expect that the closed nanopore still carries a rather large ionic current

Answer:

Thanks for your comment. This was explained in the next line – ie “The small residual current of LGC-C might stem from ion leakage either through the periphery of the pore (at the DNA duplex/lipid interface)⁵⁰⁻⁵² or through the DNA duplexes of the lid, or a combination of both.”

The closed lid nanopore conductance is almost 10 times smaller than the no lid nanopore (0.64 nS and 5.9 nS respectively). Ionic leak currents through toroidal pore formation and DNA duplexes have been previously reported^{9,10,11,12,13}.

Comment:

6) Line 283. Please indicate the applied potential.

Answer:

Many thanks – this has now been added (Line 277)

Comment:

7) Line 333. Please indicate the sampling and filtering frequencies. From the data shown in Fig. 4, bii and biv, it appears the events are largely under sampled. Please notice that in order to have meaningful rates, the events should be oversampled (e.g. the sampling rate should be at least 50 kHz and filtered at 10 kHz). [line 493, what is the meaning for having a not Bessel filter?] Although it appears from methods sampling might be correct, the data suggest that the events were sampled at lower frequency. From Fig. b(iv) the sampling rate appears 10 kHz. Fig 4bii suggest a sampling rate of 10 kHz. With a 50 kHz sampling frequency the event is therefore ~50 μ s, rather than the ~250 μ s. Please show (and collect) data at 50 kHz sampling rate.

Answer:

Many thanks for your suggestion – experiments have now been carried out at 50 kHz sampling rates and Fig 4 has been updated accordingly.

Comment:

8) Line 353. Please indicate on which side the protein was added.

Answer:

This has now been added to the text (line 360).

Comment:

9) Line 355. I think the author refers to SI Fig 32 rather than 31. Other numbering appear

*incorrect. Please check the numbering of all the SI figures.
Please indicate the concentration of trypsin used.*

Answer:

Reference to supplementary figure numbers (not supplementary document sections) are all correct.

Concentration of trypsin has now been added to the text (Line 360).

Comment:

10) Line 368. Please define what is 3.8% and 18%.

Answer:

This has now been added to the text (Line 364).

Comment:

11) Line 371. Please indicate the concentration of GFP.

Answer:

This has now been added to the text (line 379).

References

- (1) Chiari, M.; Ceriotti, L. Capillary Electrophoresis. *Food Toxicants Anal. Tech. Strateg. Dev.* **2014**, *1997* (2), 561–597.
- (2) Tan, S. J.; Campolongo, M. J.; Luo, D.; Cheng, W. Building Plasmonic Nanostructures with DNA. *Nat. Nanotechnol.* **2011**, *6* (5), 268–276. <https://doi.org/10.1038/nnano.2011.49>.
- (3) Thacker, V. V.; Herrmann, L. O.; Sigle, D. O.; Zhang, T.; Liedl, T.; Baumberg, J. J.; Keyser, U. F. DNA Origami Based Assembly of Gold Nanoparticle Dimers for Surface-Enhanced Raman Scattering. *Nat. Commun.* **2014**, *5*, 1–7. <https://doi.org/10.1038/ncomms4448>.
- (4) Göpfrich, K.; Li, C. Y.; Ricci, M.; Bhamidimarri, S. P.; Yoo, J.; Gyenes, B.; Ohmann, A.; Winterhalter, M.; Aksimentiev, A.; Keyser, U. F. Large-Conductance Transmembrane Porin Made from DNA Origami. *ACS Nano* **2016**, *10* (9), 8207–8214. <https://doi.org/10.1021/acsnano.6b03759>.
- (5) Wang, J.; Ma, J.; Ni, Z.; Zhang, L.; Hu, G. Effects of Access Resistance on the Resistive-Pulse Caused by Translocating of a Nanoparticle through a Nanopore. *RSC Adv.* **2014**, *4* (15), 7601–7610. <https://doi.org/10.1039/c3ra46032k>.
- (6) Hyun, C.; Rollings, R.; Li, J. Probing Access Resistance of Solid-State Nanopores with a Scanning Probe Microscope Tip. *Small.* **2012**, *8* (3), 385–392. <https://doi.org/10.1002/sml.201101337>. Probing.
- (7) Diederichs, T.; Pugh, G.; Dorey, A.; Xing, Y.; Burns, J. R.; Hung Nguyen, Q.; Tornow, M.; Tampé, R.; Howorka, S. Synthetic Protein-Conductive Membrane Nanopores Built with DNA. *Nat. Commun.* **2019**, *10* (1), 1–11. <https://doi.org/10.1038/s41467-019-12639-y>.
- (8) Mitscha-Baude, G.; Stadlbauer, B.; Howorka, S.; Heitzinger, C. Protein Transport through Nanopores Illuminated by Long-Time-Scale Simulations. *ACS Nano* **2021**, *15* (6), 9900–9912. <https://doi.org/10.1021/acsnano.1c01078>.
- (9) Burns, J. R.; Seifert, A.; Fertig, N.; Howorka, S. A Biomimetic DNA-Based Channel for the Ligand-Controlled Transport of Charged Molecular Cargo across a Biological Membrane. *Nat. Nanotechnol.* **2016**, *11* (2), 152–156. <https://doi.org/10.1038/nnano.2015.279>.
- (10) Göpfrich, K.; Li, C. Y.; Mames, I.; Bhamidimarri, S. P.; Ricci, M.; Yoo, J.; Mames, A.; Ohmann, A.; Winterhalter, M.; Stulz, E.; Aksimentiev, A.; Keyser, U. F. Ion Channels Made from a Single Membrane-Spanning DNA Duplex. *Nano Lett.* **2016**, *16* (7), 4665–4669. <https://doi.org/10.1021/acsnano.6b02039>.
- (11) Joshi, H.; Maiti, P. K. Structure and Electrical Properties of DNA Nanotubes Embedded in Lipid Bilayer Membranes. *Nucleic Acids Res.* **2018**, *46* (5), 2234–2242. <https://doi.org/10.1093/nar/gkx1078>.
- (12) Li, C. Y.; Hemmig, E. A.; Kong, J.; Yoo, J.; Hernández-Ainsa, S.; Keyser, U. F.;

Aksimentiev, A. Ionic Conductivity, Structural Deformation, and Programmable Anisotropy of DNA Origami in Electric Field. *ACS Nano* **2015**, *9* (2), 1420–1433. <https://doi.org/10.1021/nn505825z>.

- (13) Yoo, J.; Aksimentiev, A. Molecular Dynamics of Membrane-Spanning DNA Channels: Conductance Mechanism, Electro-Osmotic Transport, and Mechanical Gating. *J. Phys. Chem. Lett.* **2015**, *6* (23), 4680–4687. <https://doi.org/10.1021/acs.jpcllett.5b01964>.